# Learning Implicit Representation for Reconstructing Articulated Objects

**Hao Zhang, Fang Li, Samyak Rawlekar & Narendra Ahuja**
Department of Electrical and Computer Engineering
University of Illinois Urbana-Champaign
{haoz19, fangli3, samyakr2,n-ahuja}@illinois.edu

## Abstract

3D Reconstruction of moving articulated objects without additional information about object structure is a challenging problem. Current methods overcome such challenges by employing category-specific skeletal models. Consequently, they do not generalize well to articulated objects in the wild. We treat an articulated object as an unknown, semi-rigid skeletal structure surrounded by nonrigid material (e.g., skin). Our method simultaneously estimates the visible (explicit) representation (3D shapes, colors, camera parameters) and the implicit skeletal representation, from motion cues in the object video without 3D supervision. Our implicit representation consists of four parts. (1) Skeleton, which specifies how semi-rigid parts are connected. (2) Skinning Weights, which associates each surface vertex with semi-rigid parts with probability. (3) Rigidity Coefficients, specifying the articulation of the local surface. (4) Time-Varying Transformations, which specify the skeletal motion and surface deformation parameters. We introduce an algorithm that uses physical constraints as regularization terms and iteratively estimates both implicit and explicit representations. Our method is category-agnostic, thus eliminating the need for category-specific skeletons, we show that our method outperforms state-of-the-art across standard video datasets. The code is available on GitHub at: https://github.com/haoz19/LIMR. [1]

## 1 Introduction

Given one or more monocular videos as input, our goal is to reconstruct the 3D shape of the object in motion within these videos. The object in the video is articulated and exhibits two distinct types of movement: (1) Skeletal motion, which arises from the movement of its articulated bones, and (2) Surface deformation, which arises from the movement of the object's surface, such as human skin. Skeleton shape is easily represented by bone parameters, joints, and skeletal motion by their movements. Surface shape is usually represented by an irregular grid of 3D vertices placed along the surface, and the deformation by the movement of the vertices.

Certain methods Pumarola et al. (2021) learn vertex deformations over time without differentiating the two distinct movement types, resulting in high computational costs and non-smooth transitions in motion estimates. Subsequent methods Yang et al. (2021a;b; 2022); Lewis et al. (2023); Kulkarni et al. (2020) do separate the two motions, yielding better computational efficiency and smoother motion estimates. They use the Blend Skinning technique, whose efficiency hinges on the initial positions of the bones. They attempt to select better initial estimates by associating bones with groups of nearby vertices, estimated by applying K-means-clustering on the mesh vertices. Not being physically valid, this representation still yields non-intuitive bone estimates. Methods like HumanNeRF Weng et al. (2022) and RAC Yang et al. (2023) use morphological information to boost the 3D reconstruction using category-specific skeletons and pose models, which restricts the ability to autonomously acquire structural knowledge from motion.

To address the above issues, we introduce LIMR (Learning implicit Representation), a method to model not only the visible (explicit) surface shape, and color, but also the implicit sources, by

---

[1]The support of the Office of Naval Research under grant N00014-20-1-2444 and of USDA National Institute of Food and Agriculture under grant 2020-67021-32799/1024178 are gratefully acknowledged.

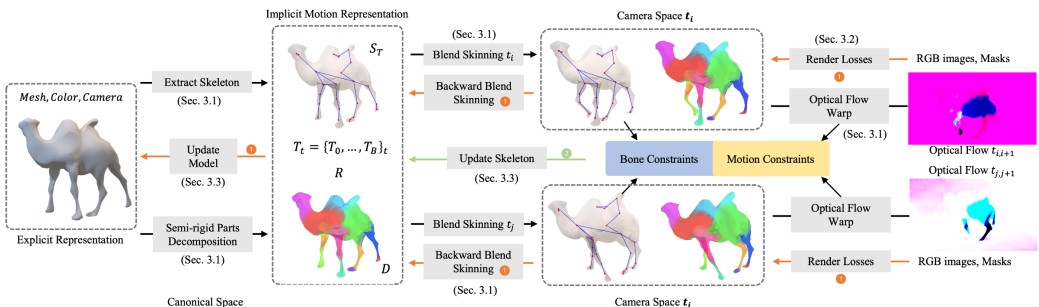

Figure 1: **Method Overview.** LIMR optimizes both the explicit representations $\mathcal{R}_e$, e.g., surface mesh, color $\mathbf{M}$, and camera parameters $\mathbf{P}_C$, implicit representation $\mathcal{R}_i$, e.g., skeleton $\mathbf{S_T}$, skinning weights $\mathbf{W}$, rigidity coefficients $\mathbf{R}$ derived from $\mathbf{W}$, and time-varying transformation $\mathbf{T}_t$ for root body and semi-rigid parts in time $t$, in an iterative manner. We optimize $\mathcal{R}_e$ using differentiable rendering frameworks (3.2, ←), and optimize $\mathcal{R}_i$ using physical constraints (3.1, ←). We optimize $\mathcal{R}_i$ and $\mathcal{R}_e$ using the SIOS$^2$ algorithm (3.3).

estimating the skeleton, its semi-rigid parts, their motions, and the articulated motion parameters given by rigidity coefficients. Our method and results show that the information present in the video cues allows such decomposition.

LIMR models the above two types of motion by learning two corresponding representations: explicit and implicit. Similar to the EM algorithm we introduce the Synergistic Iterative Optimization of Shape and Skeleton (SIOS$^2$) algorithm to learn both representations iteratively. During the Expectation (E) phase, we update the implicit skeleton by employing our current explicit reconstruction model to calculate the 2D motion direction for each semi-rigid part (bone), as well as measure the distances between connected joints at the ends of a bone. Then we update the skeleton using two physical constraints: (1) The direction of the optical flow should be similar within each semi-rigid part, and (2) distances between connected joints (bone length) should remain constant across frames. Following this, in the Maximization (M) phase, we optimize our 3D model using the updated skeleton.

The **main contributions** of this paper are as follows: **(1)** To the best of our knowledge, LIMR is the first to learn implicit representation from one (or more) RGB videos and leverage it for improving 3D reconstruction. **(2)** This shape improvement is achieved by obtaining a skeleton that consists of bone-like structures like a physical skeleton (although is not truly one), skinning weights, and rigidity coefficients. **(3)** Along with the implicit representation, our SIOS$^2$ algorithm also synergistically optimizes the explicit representation. **(4)** Because LIMR derives its estimates without any prior knowledge of object shape, it is category-agnostic. **(5)** Experiments with a number of standard videos show that LIMR improves 3D reconstruction performance with respect to 2D Keypoint Transfer Accuracy and 3D Chamfer Distance in the range of 3%-8.3% and 7.9%-14% over state-of-the-art methods.

## 2 RELATED WORK

**Template-Guided and Class-Specific Reconstructions.** Various 3D template methods are employed differently: some deform template vertices Zuffi et al. (2018), others segment the mesh, operate on these parts, and reassemble for reconstruction Zuffi et al. (2017); Xiang et al. (2019). Some use efficient vertex deformation through blend skinning Wang & Phillips (2002); Kavan et al. (2007). Some introduce 3D poses, learning joint angles with ground truth data for better reconstruction Zuffi et al. (2019); Biggs et al. (2020); Badger et al. (2020); Kocabas et al. (2020). These approaches excel with known object categories and abundant 3D data but struggle with limited 3D data or unknown categories. RAC Yang et al. (2023) recently introduced category-specific shape models, yielding good 3D reconstruction. Current trends aim to minimize reliance on 3D annotations, opting for 2D annotations like silhouettes and key points Goel et al. (2020); Kanazawa et al. (2018); Li et al. (2020b). Single-view image reconstruction achieves impressive results without explicit 3D annotations Li et al. (2020a); Kanazawa et al. (2018); Kulkarni et al. (2020). MagicPony

Wu et al. (2023b) learns 3D models for objects such as horses and birds from single-view images, but faces challenges with fine details and substantial deformations, especially for unfamiliar objects.

**Class-Independent and Template free methods for Video-Based Reconstruction.** Nonrigid structure from motion (NRSfM) does not rely on 3D data or annotations. It utilizes off-the-shelf 2D key points and optical flow for videos, regardless of the category Teed & Deng (2020). Despite handling generic shapes, NRSfM requires consistent long-term point tracking, which isn't always available. Recently, neural networks have learned 3D structural details from 2D annotations for specific categories. However, they struggle with long-range and rapid motion, particularly in uncontrolled video settings. LASR and ViSER Yang et al. (2021a;b) reconstruct articulated shapes from monocular videos using differentiable renderingLiu et al. (2019), albeit sometimes yielding blurry geometry and unrealistic articulations. Banmo Yang et al. (2022) addresses this by leveraging numerous frames from multiple videos to generate more plausible results. However, obtaining such a diverse collection of videos with varying viewpoints can be challenging.

**Neural Radiance Fields.** In scenarios with registered camera poses, NeRF and its variations Wang et al. (2021b); Jeong et al. (2021); Lin et al. (2021); Wang et al. (2021b); Li et al. (2023) typically optimize a continuous volumetric scene function to synthesize novel views within static scenes containing rigid objects. Some approaches Pumarola et al. (2021); Li et al. (2021); Park et al. (2021a;b); Tretschk et al. (2021) attempt to handle dynamic scenes by introducing new functions to transform time-varying points into a canonical space. However, they struggle when confronted with significant relative motion from the background. To address these challenges, pose-controllable NeRFs Wu et al. (2022); Liu et al. (2021) have been proposed, but they either heavily rely on predefined category-level 3D data or synchronized multi-view videos. In our method, we refrain from using any provided ground truth 3D information during optimization and can generate more 3D information compared to other techniques.

## 3 METHOD

Given one or more videos of articulated objects, our method learns explicit representations $\mathcal{R}_e = \{\mathbf{M}, \mathbf{P}_c^t\}$, as well as implicit representation $\mathcal{R}_i = \{\mathbf{S_T}, \mathbf{W}, \mathbf{R}, \mathbf{T}\}$ synergistically. $\mathbf{M}$ represents the surface mesh and color in the object-centered (canonical) space, $\mathbf{P}_c^t$ are camera parameters in frame $t$. The implicit representation includes the skeleton $\mathbf{S_T} = \{\mathbf{B} \in \mathbb{R}^{B \times 13}, \mathbf{J} \in \mathbb{R}^{J \times 5}\}$, the skinning weights $\mathbf{W} \in \mathbb{R}^{N \times B}$, the vertices-wise rigid coefficients $\mathbf{R} \in \mathbb{R}^E$, and the time-varying transformations of root body and $B$ semi-rigid parts $\mathbf{T}^t = \{\mathbf{T}_0^t, \mathbf{T}_1^t, ..., \mathbf{T}_B^t\}, \mathbf{T}_b^t \in SE(3)$. Bones $\mathbf{B}$ include the Gaussian centers ($\mathbf{C} = \mathbf{B}[:,:3]$), precision matrices ($\mathbf{Q} = \mathbf{B}[:,3:12]$), and lengths of bones ($\mathbf{B}[:,12]$). $\mathbf{J}$ denotes the joints, containing the indices of two connected bones ($\mathbf{J}[:,:2]$) and the coordinates of the joint ($\mathbf{J}[:,2:]$). $B$, $E$, and $J$ are the numbers of bones, edges between vertices, and joints. The implicit representations are optimized using physical constraints such as bone length being consistent and optical flow directions being similar in the same semi-rigid parts across time (Sec. 3.1) The implicit representations are optimized using physical constraints such as bone length being consistent and optical flow directions being similar in the same semi-rigid parts across time (Sec. 3.1), and the explicit representations are optimized using differentiable rendering (Sec. 3.2). We leverage the Synergistic Iterative Optimization of Shape and Skeleton (SIOS$^2$) algorithm to optimize both representations in an iterative manner (Sec. 3.3).

### 3.1 IMPLICIT REPRESENTATION LEARNING

**Skeleton Initialization by Mesh Contraction.** Given a mesh $\mathbf{M} = \{\mathbf{X}, \mathbf{E}\}$, where $\mathbf{X}, \mathbf{E}$ are vertices and edges, instead of using K-means to cluster centers from vertices, we use a **Laplacian Contraction** Cao et al. (2010) to obtain the initial skeleton $\mathbf{S_T}$. It starts by contracting the mesh geometry into a zero-volume skeletal shape using implicit Laplacian smoothing with global positional constraints; details are given in Sec.A.1. This iterative contraction captures essential features, usually in fewer than ten iterations, with convergence when the mesh volume approaches zero. This contraction process retains key features of the original mesh and does not alter connectivity. The contracted mesh is then converted into a 1D skeleton through the **connectivity surgery** process of Au et al. (2008) that removes all collapsed faces while preserving the shape and topology of the contracted mesh. In the process of connectivity surgery on a contracted 2D mesh, edge collapses are driven by a dual-component cost function: a shape term and a sampling term. The shape cost,

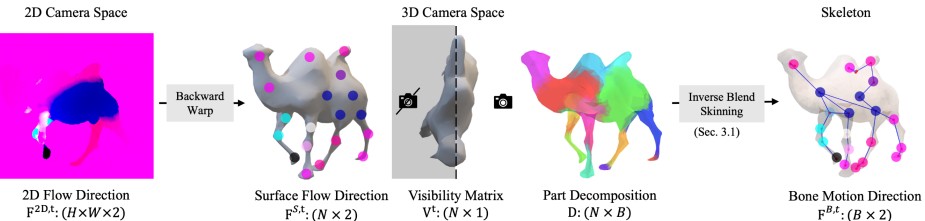

Figure 2: **Optical Flow Warp.** We backward project the 2D optical flow to the camera space, obtaining flow direction $\mathbf{F}^{S,t}$ for every vertex on the surface and calculate the visibility matrix $\mathbf{V}$ according to the viewpoint. Then we apply inverse blend skinning with $\mathbf{F}^{2D,t}$, $\mathbf{V}$ and skinning weights $\mathbf{W}$ as inputs to calculate the bone motion direction $\mathbf{F}^{B,t}$. Note $t$ denotes mapping from frame $t$ to $t+1$.

inspired by the QEM simplification method Garland & Heckbert (1997), aims to retain the inherent geometry of the original mesh during simplification, ensuring minimal shape distortion. Conversely, the sampling cost is tailored to deter the formation of disproportionately long edges by assessing the cumulative distance traveled by adjacent edges during an edge collapse. This consideration prevents over-simplification in straight mesh regions, preserving a granular and true representation of the mesh's original structure.

**Skinning Weights.** For given vertices on the surface, we define a soft skinning weights matrix $\mathbf{W} \in \mathbb{R}^{N \times B}$, which assigns $N$ vertices to $B$ bones probabilistically. Optimization may be difficult for learning such a matrix. Therefore, similar to BANMo, we obtain the decomposition matrix by calculating the Mahalanobis distance between surface vertices and $B$ bones as follows:

$$\mathbf{W}_{n,b} = \text{softmax}(d(\mathbf{X}_n, \mathbf{C}_b, \mathbf{Q}_b) + \mathbf{MLP_W}(\mathbf{X}_n)),$$
$$d(\mathbf{X}_n, \mathbf{C}_b, \mathbf{Q}_b) = (\mathbf{X}_n - \mathbf{C}_b)^T \mathbf{Q}_b (\mathbf{X}_n - \mathbf{C}_b), \quad \mathbf{Q}_b = \mathbf{V}_b^T \mathbf{\Lambda}_b \mathbf{V}_b, \tag{1}$$

where $d$ is the distance function between vertex $n$ and bone $b$. Each bone is defined as a Gaussian ellipsoid, which contains three learnable parameters: center $\mathbf{C} \in \mathbb{R}^B \times 3$, orientation $\mathbf{V} \in \mathbb{R}^{B \times 3 \times 3}$, and diagonal scale $\mathbf{\Lambda} \in \mathbb{R}^{B \times 3 \times 3}$ and $\mathbf{X}_n$ denotes the coordinates of a vertex n.

**Rigidity Coefficient.** A significant factor in representing deformations within an articulated object is the degrees of freedom each vertex possesses relative to its neighboring vertices on the surface. To systematically account for this, we introduce a rigidity coefficient matrix, $\mathbf{R}$, with dimensions $\mathbb{R}^E$, where $E$ denotes the total edge count linking the vertices. It is important to recognize that regions around joint areas are inherently more pliable and exhibit pronounced deformations. This is in stark contrast to vertices anchored in the midst areas of the semi-rigid components. Such differential susceptibility leads to observable correlated movements amongst proximate vertices. For a given skeleton, represented by $\mathbf{S_T}$, and the corresponding semi-rigid decomposition $\mathbf{W}$ of the object, the coefficient matrix $\mathbf{R}$ is formulated by computing the product of entropies from the probability distributions of two connected vertices assigned to $B$ bones:

$$\mathbf{R}_{i,j} = \left( \sum_{b=0}^{B-1} \mathbf{W}_{i,b} \log_2 \mathbf{W}_{i,b} + \lambda \right)^{-1} \times \left( \sum_{b=0}^{B-1} \mathbf{W}_{j,b} \log_2 \mathbf{W}_{j,b} + \lambda \right)^{-1}. \tag{2}$$

Here, $\mathbf{R}_{i,j}$ signifies the rigidity coefficient connecting vertices $i$ and $j$. The term $\lambda$ acts as a stabilization constant to avoid division by zero, with a default setting at $0.1$. We restrict our computations to pairs of directly linked vertices.

**Dynamic Rigid.** Capitalizing on this rigidity coefficient, we propose the DR (Dynamic Rigid) loss, serving as an improvement over the conventional ARAP (As Rigid As Possible) loss, which has been extensively adopted as a motion constraint in several prior works Yang et al. (2021a); Sumner et al. (2007); Tulsiani et al. (2020); Yang et al. (2021b). ARAP loss encourages the distance between all adjacent vertices to remain constant across continuing frames. However, this approach is not always helpful. Specifically, vertices near the joints should inherently possess more degrees of freedom compared to those located in the midst of semi-rigid parts. To address this, our proposed Dynamic Rigidity (DR) offers a significant enhancement, allowing for more natural and adaptive

spatial flexibility based on the vertex's proximity to joints:

$$\mathcal{L}_{\text{DR}} = \sum_{i=1}^{n} \sum_{j \in N_i} \mathbf{R}_{i,j} \left| \left\| \mathbf{X}_i^t - \mathbf{X}_j^t \right\|_2 - \left\| \mathbf{X}_i^{t+1} - \mathbf{X}_j^{t+1} \right\|_2 \right|, \tag{3}$$

where $\mathbf{X}_i^t$ denotes the coordinates of vertex $i$ at time $t$ and $N_i$ represents the set of neighboring vertices of vertex $i$.

**Blend Skinning.** We utilize blend skinning to map the surface vertex $\mathbf{X}_n^0$ from canonical space (time 0) to camera space time $t$: $\mathbf{X}_n^t$. Given bones $\mathbf{B}$, skinning weights $\mathbf{W}$ and time-varying transformation $\mathbf{T}^t = \{\mathbf{T}_b^t\}_{b=0}^{B}$, we have: $\mathbf{X}_n^t = \mathbf{T}_0^t(\sum_{b=1}^{B} \mathbf{W}_{n,b} \mathbf{T}_b^t)\mathbf{X}_n^0$, where each vertex is transformed by combining the weighted bone transformation $\mathbf{T_b^t}, b > 0$ and then transformed to the camera space by root body transformation $\mathbf{T}_0^t$. On the contrary, we employ the backward blending skinning operation to map vertices from the camera space to the canonical space. The sole distinction from blend skinning lies in utilizing the inverse of the transformation $\mathbf{T}$ instead of $\mathbf{T}$.

**Optical Flow Warp.** Given the 2D optical flow between times $t$ and $t + 1$, represented as $\mathbf{F}^{\text{2D},t}$, our goal is to determine the 2D motion direction for each semi-rigid component (underlying bone). This bone motion direction is denoted as $\mathbf{F}^{B,t}$ and is expressed in $\mathbb{R}^{B \times 2}$. Notably, these directions serve as a critical physical constraint in the skeleton refinement process, as outlined in Section 3.3. Intuitively, the motion of a bone is an aggregate of the motions of the vertices associated with it. To achieve this, we project each pixel from the 2D optical flow back to the 3D camera space. Due to the fact that vertices may not map perfectly onto pixels, the surface flow direction $\mathbf{F}^{S,t} \in \mathbb{R}^{N \times 2}$ is derived using bilinear interpolation. It is essential to note that attributing optical flow direction to vertices on the non-visible side of the surface is erroneous. To address this, we compute a visibility matrix $\mathcal{V}^t \in \mathbb{R}^{N \times 1}$ based on the current mesh and viewpoint of time $t$ using ray-casting. Optical flow directions corresponding to non-visible vertices are then nullified. Then, we approach the estimation of bone motion direction by employing blend skinning, which coherently integrates the optical flow directions associated with surface vertices. Specifically, given a defined skinning weights, represented as $\mathbf{W}$, and the computed surface flow direction $\mathbf{F}^{S,t}$, the bone motion direction can be represented as:

$$\mathbf{F}_b^{B,t} = \sum_{n=0}^{N-1} \mathbf{W}_{n,b} \mathbf{F}_n^{S,t} \mathcal{V}_n^t, \quad \mathbf{F}^{B,t} = \{\mathbf{F}_0^{B,t}, ..., \mathbf{F}_{B-1}^{B,t}\} \tag{4}$$

In the above equation, $\mathbf{F}_b^{B,t}$ signifies the flow direction associated with bone $b$, while $\mathbf{F}_n^{S,t}$ and $\mathcal{V}_n^t$ denote the flow direction and visibility status for vertex $n$, respectively.

**Joint Localization & Bone Length Calculation.** Given the current bone coordinates $\mathbf{B}$, bone connections $\mathbf{J}[:, : 2]$, and the skinning weights $\mathbf{W}$, we first identify the set of vertices $V_{i,j}$ that lie above the joint $\mathbf{J}_{i,j}$ connecting bones $i$ and $j$. A vertex $v_n$ is included in $V_{i,j}$ if both $\mathbf{W}_{n,i}$ and $\mathbf{W}_{n,j}$ are greater than or equal to $t_r$ (default by 0.4). The coordinates of $\mathbf{J}_{i,j}$ are then computed as the mean coordinates of the vertices in $V_{i,j}$. Specifically, $\mathbf{J}_{i,j}[:, 2 :] = \frac{\sum_{v_n \in V_{i,j}} \mathbf{X}_n}{|V_{i,j}|}$, where $\mathbf{X}_n$ represents the coordinates of $v_n$. Once we have the coordinates for each joint, the length of each bone is calculated as the distance between the coordinates of the corresponding joint pairs.

## 3.2 EXPLICIT REPRESENTATIONS MODEL

There have been two main approaches to learning explicit representations ($\mathcal{R}_e$): NeRF-based approach Wang et al. (2021a); Pumarola et al. (2021), and the neural mesh renderer as described in soft rasterizer Liu et al. (2019).

**Losses & Regularizations.** Similar to BANMo, the Nerf-based approaches use reconstruction loss, feature loss, and a regularization term: $\mathcal{L}_{\text{NeRF-based}} = \mathcal{L}_{\text{reconstruction}} + \mathcal{L}_{\text{feature}} + \mathcal{L}_{\text{3D-consistency}}$ The reconstruction loss follows the existing differentiable rendering pipelines Yariv et al. (2020); Mildenhall et al. (2021), which is the sum of $\mathcal{L}_{\text{silhouette}}$, $\mathcal{L}_{\text{RGB}}$, and $\mathcal{L}_{\text{Optical-Flow}}$. The feature loss is composed of the 3D feature embedding loss ($\mathcal{L}_{\text{feature-embedding}}$) by minimizing the difference between the canonical embeddings from prediction and backward warping. The 3D consistency loss ($\mathcal{L}_{\text{3D-consistency}}$) from NSFF Li et al. (2021) is used to ensure the forward-deformed 3D points match their original location upon backward-deformation. A.2.

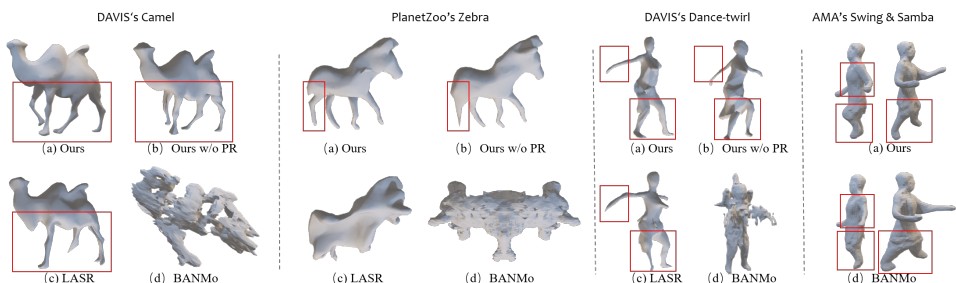

Figure 3: **Mesh Results.** We show the reconstruction results of (a) Our approach, (b) Our approach w/o part refinement, (c) LASR, and (d) BANMo in the DAVIS's `camel`, `dance-twirl` and PlanetZoo's `zebra`.

The loss function of Neural Mesh Rasterization-based methods includes a reconstruction loss, motion, and shape regularization: $\mathcal{L}_{\text{NMR-based}} = \mathcal{L}_{\text{Reconstruction + perceptual}} + \alpha \times \mathcal{L}_{\text{shape}} + \eta \times \mathcal{L}_{\text{DR}}$ where $\alpha$ and $\eta$ are set the same as LASR. The reconstruction loss is the same as the NeRF-based, with the addition of perceptual loss Zhang et al. (2018). Furthermore, we use shape and motion regularizations, and the soft-symmetry constraints. We replace ARAP loss with **Dynamic Rigid** (DR) loss to encourage the distance between two connected vertices, within the same semi-rigid part, to remain constant. The Laplacian operator is applied to generate a smooth surface ($\mathcal{L}_{\text{shape}}$). The details are presented in Appendix A.2.

**Part Refinement.** For the NMR-based scheme, the silhouette loss contributes the most to the rendering loss. For objects like quadrupeds whose thin limbs only occupy a small part of the 2D mask, the contribution of limbs to the overall silhouette loss is relatively less. This phenomenon always results in a well-reconstructed torso, but bad-reconstructed limbs. So with the skinning weights $\mathbf{W} \in \mathbb{R}^{N \times B}$ obtained in Sec.3.1, we then train only the limb parts by freezing all parameters but those for limbs. The selection process is given by $\mathbf{W}_{\text{one-hot}}[i,:] = \mathbf{W}[\mathbf{W}[i,:] == \arg\max_i \mathbf{W}[I,:]]$ and $\mathbf{W}_{\text{one-hot}} \in \{0,1\}^{N \times B}$ represent the one-hot encoded decomposition matrix, where each vertex is assigned to exactly one part. Given $\mathbf{W}_{\text{one-hot}}$, we update the vertices only from small parts.

## 3.3 Synergistic Iterative Optimization

Our approach is grounded on two interdependent sets of learnable parameters. The first set is associated with the reconstruction model, comprising the mesh, color, camera parameters, and time-varying transformations. Simultaneously, the secondary set pertains to the underlying skeleton, encompassing the bones and articulation joints. To achieve a harmonized optimization of these parameter sets, we utilize the Synergistic Iterative Optimization of Shape and Skeleton (SIOS$^2$) algorithm, as elucidated in Alg.1. This algorithm updates both parameter sets iteratively similar to the EM algorithm. During the **E-step**, the skeleton remains fixed and is utilized in the blend skinning operation and regularization losses. The reconstruction model is then updated based on the rendering losses and regularization components. Subsequently, using the updated reconstruction model, we determine the bone motion direction and bone lengths for the selected frames. This information is utilized in refining the skeleton in light of physical constraints.

**Skeleton Refinement.** In the **M-step**, the skeleton is updated using two constraints: (1) Merge bones $\mathbf{B}_b, \mathbf{B}_{b'}$ if they consistently move in sync across $\mathbf{H}$ selected images $\{\mathbf{I}_f\}_{f=0}^{\mathbf{H}}$, indicating they belong to a single semi-rigid part. (2) Introduce a joint between $\mathbf{J}_j, \mathbf{J}_{j'}$ if their distances fluctuate significantly across specific frames. Given that,

$$\text{if } \max_f \text{Length}(\mathbf{B}_{b'}^f) - \min_f \text{Length}(\mathbf{B}_{b'}^f) > t_d \qquad \Rightarrow \widetilde{\mathbf{J}} \text{ introduced;}$$

$$\text{if } \min_f \mathcal{S}(\mathbf{F}_B^{b',f}, \mathbf{F}_B^{b'',f}) > t_o \qquad \Rightarrow \mathbf{B}_{b'} \text{ and } \mathbf{B}_{b''} \text{ merged to form } \widetilde{\mathbf{B}},$$

where $\mathcal{S}(\mathbf{a}, \mathbf{b}) = \frac{\mathbf{a} \cdot \mathbf{b}}{||\mathbf{a}||_2 \times ||\mathbf{b}||_2}$ is the cosine similarity. Furthermore, the coordinate of $\widetilde{\mathbf{J}}$ is computed by the mean of $\mathbf{J}_j$ and $\mathbf{J}_{j'}$ and we remove the connection between them and connect them to $\widetilde{\mathbf{J}}$. $\widetilde{\mathbf{B}} = w_b' \mathbf{B}_{b'} + w_{b''} \mathbf{B}_{b''}$, with $w_b' = \frac{\exp \sum_{b=b'} \mathbf{W}_{n,b}}{\exp \sum_{b=b'} \mathbf{W}_{n,b} + \exp \sum_{b=b''} \mathbf{W}_{n,b}}$, which is applied on Gaussian centers

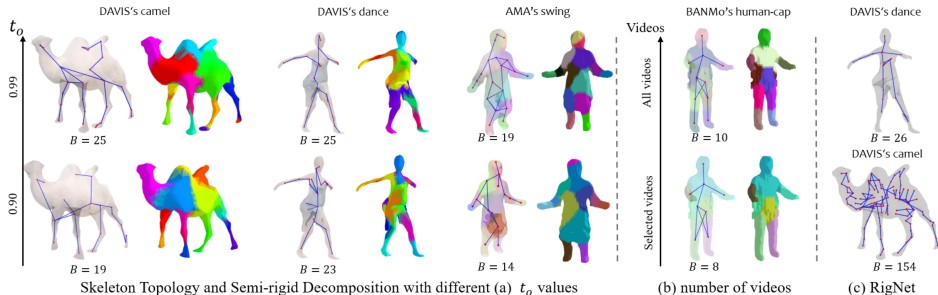

Figure 4: **Implicit Representation Results**: (a) variations with different $t_o$ values. (b) from different videos. From left to right are the results for DAVIS's `camel`, `dance-twirl`, AMA's `swing`, and BANMo's `human-cap`. (c) shows the skeleton generated by RigNet Xu et al. (2020). $B$ indicates the number of bones.

**C** and precision matrices **Q** but the new bone length is calculated directly from the coordinates of the joints at the far end of two bones.

## 4 EXPERIMENTAL RESULTS

In this section, we evaluate our approach with a variety of experiments. The experimental evaluation is divided into two scenarios: 3D reconstruction leveraging (1) single monocular short videos, and (2) videos spanning multiple perspectives. We select the state-of-the-art methods LASR Yang et al. (2021a) and BANMo Yang et al. (2022) as our baselines for the two respective scenarios mentioned above. For a fair comparison, we adopt NeRF-Based and NMR-Based explicit representation models when compared with BANMo and LASR respectively. Additional details and results for the benchmarks and implementation can be found in the supplementary material, and we plan to make the source code publicly available.

### 4.1 RECONSTRUCTION FROM SINGLE VIDEO

Compared to using multiple videos, reconstructing from a single short monocular video is evidently more challenging. Hence, we aim to verify if our proposed implicit representation can achieve robust representation under such data-limited conditions. Firstly, we tested our approach on the well-established BADJA Biggs et al. (2019) benchmark derived from DAVIS Perazzi et al. (2016) dataset. Additionally, we broadened our experimental scope by introducing the PlanetZoo dataset manually collected from YouTube. The Plantzoo dataset amplifies a more notable challenge than the DAVIS dataset on the more extensive camera motion of each video contributing to the biggest distinction. Datasets details are provided in Sec.A.4.

**Qualitative Comparisons.** We present a comparative analysis of mesh reconstruction with LIMR, LASR, and BANMo, illustrated in Fig.3. With single monocular video, e.g. DAVIS's `camel`, `dance-twirl` and PlanetZoo's `zebra`, BANMo does not generate satisfactory results as depicted in (d) due to the constraints of NeRF-based explicit representation models, demanding massive input video and adequate viewpoints. Conversely, NMR-based methodologies like LASR exhibit acceptable results. However, the absence of an understanding of articulated configurations of objects induces notable inaccuracies. For instance in Fig.3 `camel`, LASR erroneously elongates vertices beneath the abdominal region and conflates the posterior extremities to align the rendered mask with the ground truth mask, thus falling into an incorrect local optimum. LIMR, as shown in Fig.3 (a), learns the implicit representation which helps it overcome problems like those seen with LASR. Our approach steers the canonical mesh with the learned skeleton, ensuring alignment with precise skeletal dynamics and satisfactory reconstruction outcomes. LIMR is lenient with regard to video and view requisites. It learns a skeleton from even a single monocular video.

**Quantitative Comparisons.** Table 1 shows the performance of LIMR concerning 2D key points transfer accuracy against state-of-the-art methods: LASR and ViSER Yang et al. (2021b), spanning both DAVIS and PlanetZoo datasets. While LASR and ViSER have their respective strengths depending on the videos, LIMR consistently outperforms both of them across all tested videos.

Table 1: 2D Keypoint transfer accuracy on DAVIS and PlanetZoo videos.

| Method | DAVIS | | | | | | PlanetZoo | | | | | |
|---|---|---|---|---|---|---|---|---|---|---|---|---|
| | camel | dog | cow | bear | dance | Ave. | dog | zebra | elephant | bear | dinosaur | Ave. |
| ViSER | 76.7 | 65.1 | 77.6 | 72.7 | 78.3 | 74.1 | - | - | - | - | - | - |
| LASR | 78.3 | 60.3 | 82.5 | 83.1 | 55.3 | 71.9 | 73.4 | 57.4 | 69.5 | 63.1 | 71.3 | 66.9 |
| Ours w/o DR | 79.1 | 65.4 | 83.2 | 85.3 | 75.9 | 78.5 | 74.7 | 58.3 | 70.1 | 64.9 | 72.8 | 68.2 |
| Ours | **80.3** | **67.1** | **83.4** | **86.8** | **83.1** | **80.2** | **77.5** | **61.1** | **70.9** | **66.6** | **73.6** | **69.9** |

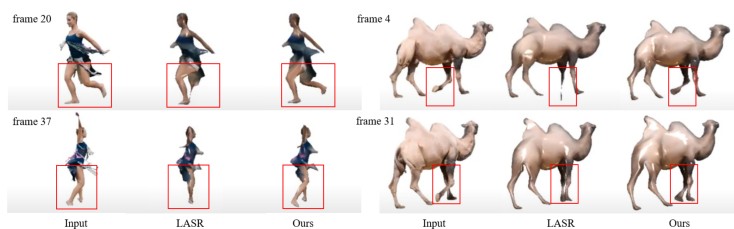

Figure 5: **Rendering Results.** Compare the rendering results on DAVIS's `camel,dance-twirl` with prior art LASR.

Specifically, within the DAVIS dataset, our approach surpasses LASR by $8.3\%$ and ViSER by $6.1\%$ and achieves a $3\%$ advantage over LASR on the PlanetZoo dataset. Note that ViSER is sensitive to large camera movement, and a large number of input frames, which causes ViSER to perform badly in the PlanetZoo. Given BANMo's inability to produce commendable outcomes from a single short monocular video, we did not compare LIMR with BANMo.

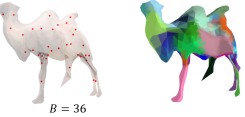

Figure 6: Bone localization and part decomposition results from LASR.

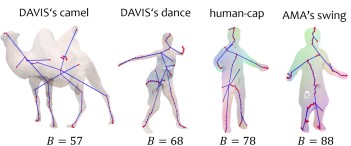

Figure 7: Initial skeleton obtained via mesh contraction.

## 4.2 RECONSTRUCTION FROM MULTIPLE VIDEOS

For evaluating the reconstruction with multiple video inputs, we conducted experiments on BANMo's `cat`, `human-cap`, AMA's `swing`, and `samba` datasets. As illustrated in Fig.3 `swing` (b), compared with BANMo, our method further refines skeletal motion through the assimilation of implicit representations. Through observation, compared to BANMo, our method can achieve more accurate reconstruction results in areas with larger motion ranges, such as the knees and elbows. Notably, BANMo requires a pre-defined number of bones, which is defaulted by $25$. In contrast, we learn the most suitable number of bones adaptively. In the `swing` experiment, we achieved better results than BANMo using only $19$ bones rather than $25$. Also, quantitative results of LIMR on AMA's `Swing` and `Samba` compared with BANMo are shown in Tab.3. Under equivalent training conditions, our method employs fewer bones yet outperforms BANMo by around $12\%$.

## 4.3 DIAGNOSTICS

Here we ablate the importance of each component, show the outcomes with different training settings, and analyze why our method outperforms the existing methods.

**Physical-Like Skeleton vs Virtual Bones.** The key distinction of our approach from existing methods is our pursuit to learn a skeleton for articulated objects, in contrast to the commonly-used virtual bones. As demonstrated in Fig.6, LASR tends to concentrate bones in the torso area, allocating only a few bones to the limbs. However, in practice, limbs often undergo more significant skeletal motion compared to the torso. This discrepancy causes existing methods to underperform, particularly at limb joints. In Fig.6 the entire limb of the camel has just one bone, which is treated as a semi-rigid part, and prevents bending. Rendering results in Fig.5, 8 from the `dance-twirl`, `camel`, `dog`, and `zebra` experiments show our better limb reconstruction compared to LASR. Similarly, in `swing` experiment in Fig.3, LIMR delivers better results in the knee region, surpassing BANMo.

**Skeleton Comparision.** Firstly, we attempt to define the conditions necessary for an effective skeleton in the task of 3D dynamic articulated object reconstruction: (1) The distribution of bones should

be logical and tailored to the object's movement complexity as discussed in the above paragraph (2) The skeleton must be detailed enough to accurately represent the object's structure while avoiding excessive complexity that might arise from local irregularities. As shown in Fig.4 we compared the skeleton results from RigNet Xu et al. (2020) (c) with our skeleton (a) for DAVIS-`camel`, `dance`. Despite RigNet employing 3D supervision, including 3D meshes, ground truth skinning weights, and skeletons, it fails to achieve better skeleton results than LIMR, which operates without any 3D supervision. For instance, RigNet tends to assign an excessive number of unnecessary bones to areas with minimal motion, such as the torso of a camel. Additionally, in the results for `dance`, it lacks skeletal structure in crucial areas like the right lower leg and the left foot.

**Skeleton Initialization.** We learn the skeleton by first leveraging mesh contraction to get an initial skeleton with a larger number of bones and then updating the skeleton according to the $SIOS^2$ algorithm. As shown in Fig.7, the initial skeletons always contain outlier points due to the unsmooth mesh surface, while our algorithm can effectively remove such noisy points during training.

**Different Thresholds for skeleton Refinement.** Throughout the skeleton updating process, we observed a notable stability of our results vs minor variations in the threshold $t_d$ around $0.5\times$ current bone length, but a pronounced sensitivity to changes in $t_o$. As depicted in Fig.4 (a), LIMR tends to retain more bones and a more complex skeleton with higher $t_o$. This also leads to more semi-rigid parts in decomposition. Empirically, we observed marginal differences in reconstruction results when $t_o$ lies between 0.99 and 0.85. However, excessively large or diminutive threshold values lead to too many or fewer bones resulting in poor reconstructions.

**Impact of Video Content on skeleton.** Our method intrinsically derives a skeleton by relying on the motion cues presented in the input videos. Consequently, varying motion contents across videos lead to distinct skeletons. Illustratively, in Fig.4 (b), when provided with all 10 videos from `human-cap`, which includes actions within arms and legs, we learn a skeleton composed of 10 bones, allocating 2 bones for each leg. Conversely, using a selected subset of videos where leg motions are conspicuously absent, our system recognizes the leg as a semi-rigid component, assigning a single bone for each. Also, the quality of the skeleton is influenced by the content within the video. For instance, in Fig.4, the skeleton for `human-cap` appears to be of higher quality compared to `swing`. This is attributed to the diverse motions present in `human-cap`, which includes actions such as raising hands, leg movements, and squatting. Conversely, `swing` showcases a more simplistic motion.

**Efficacy of Dynamic Rigidity and Part Refinement.** We highlight the performance enhancements brought by introducing Dynamic Rigidity (DR), in contrast to the ARAP loss in contemporary works. As delineated in Tab.1, the integration of DR in our approach yields a performance boost of 1.7% on the DAVIS dataset and 1.7% improvement on the PlanetZoo dataset. Qualitative comparisons are shown in Fig.9. As illustrated in Fig.3, leveraging the learned skinning weights enables localized optimization for articulated objects, markedly enhancing the reconstruction fidelity of specific regions. Notably, the leg reconstruction outcomes in both the `camel` and `zebra` experiments witnessed substantial improvements post the part refinement procedure.

## 5   CONCLUSION & LIMITATIONS

To **conclude**, we have introduced a method to simultaneously learn explicit representations (3D shape, color, camera parameters) and implicit representations (skeleton) of a moving object from one or more videos without 3D supervision. We have proposed an algorithm $SIOS^2$ that iteratively estimates both representations. Experimental evaluations demonstrate the use of the 3D structure captured in the implicit representation enables LIMR to outperform state-of-the-art methods.

Following are some **limitations** of LIMR. (1) Since it starts with a simple sphere instead of a predefined shape template, it requires input videos to include diverse views for the best results. (2) Most of the available solutions, including LIMR, suffer from the problem of continuously moving cameras, which requires continuous estimation of the changing camera. Any errors therein propagate to errors in shape estimates since the model learns the shape and camera views simultaneously. For example, a poor shape estimate may lead to a wrong symmetry plane. We plan to improve the model's resilience to camera-related errors. (3) Our model, like many others, requires long training (10-20 hours per run on 1 A100 GPU with 40GB), and we plan on working to reduce it.

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

| Method | Shape Template | Skeleton Template | Motion Manipulation | Multiple Videos | Camera Pose | Learn Skeleton |
|---|---|---|---|---|---|---|
| LASR Yang et al. (2021a) | × | × | virtual | × | × | × |
| ViSER Yang et al. (2021b) | × | × | virtual | × | × | × |
| BANMo Yang et al. (2022) | × | × | virtual | ✓ | ✓ | × |
| CASA Wu et al. (2022) | ✓ | ✓ | physical | × | × | × |
| WIM Noguchi et al. (2022) | × | × | physical | ✓ | ✓ | ✓ |
| CAMM Kuai et al. (2023) | × | ✓ | physical | ✓ | ✓ | × |
| RAC Yang et al. (2023) | × | ✓ | physical | ✓ | ✓ | × |
| MagicPony Wu et al. (2023a) | × | ✓ | physical | ✓ | × | × |
| Ours (LIMR) | × | × | physical | × | × | ✓ |

Table 2: Difference between LIMR and existing methods

# A    APPENDIX

In this section, we provide: **(1)** More information about the mesh contraction operation in A.1. **(2)** Details of the losses and regularization used in the NeRF-based scheme (A.2.2) and NMR-based scheme (A.2.1). **(3)** Additional qualitative results (for rendering and mesh) for different videos in the PlanetZoo dataset in Fig.8 11 9 13. **(4)** Quantitative results for AMA's `swing and samba` in Tab.3. **(5)** Details of the test datasets in A.4. **(6)** Mesh results during skeleton updating in Fig.12 **(7)** Details of the SIOS$^2$ algorithm in Alg.1. **(8)** Difference between LIMR and existing methods in Tab.2 and Sec.A.6. **(9)** More discussion of the difference between two explicit representation schemes in A.5, and failure cases due to wrong camera predictions in A.5. **(10)** More discussion about our bone motion estimation strategy compared with WIM Noguchi et al. (2022). **(11)** Notations in Tab.A.5.

---

**Algorithm 1** Synergistic Iterative Optimization of Shape and Skeleton (SIOS$^2$)

---

1: Initialize the parameters $\theta^{(0)}$ for the reconstruction model and extract the initial skeleton $\mathbf{S_T}^{(0)}$ from the current rest mesh $\mathbf{M}^{(0)}$ utilizing mesh contraction (Sec.3.1).
2: **for** $e = 0, 1, 2, \ldots$ until convergence **do** ▷ E-Step
3:     Transform $\mathbf{M}^{(e)}$ to time-space $\mathbf{M}_t^{(e)}$ by integrating the transformations of bones (Sec.3.1).
4:     Render and compute the reconstruction losses (Sec.3.2).
5:     Update the parameters in the reconstruction model: $\theta^{(e+1)} \leftarrow \theta^{(e)}$ (Sec.3.3).
                                                                        ▷ M-Step
6:     Randomly sample $\mathbf{H}$ images from all frames.
7:     Compute the bone motion direction in the selected frames using optical flow warping (Sec.3.1).
8:     Determine joint coordinates and bone lengths in the selected frames (Sec.3.1).
9:     Refine the skeleton considering physical constraints: $\mathbf{S_T}^{(e+1)} \leftarrow \mathbf{S_T}^{(e)}$ (Sec.3.3).
10: **end for**

---

## A.1    MESH CONTRACTION

**Laplacian Contraction**, driven by weighted forces, creates a thin skeleton representing the object's logical components. The contracted vertex positions, $\mathbf{X}'$, are determined by a Laplace equation: $\mathbf{LX}' = 0$. Here, $\mathbf{L}$ is the curvature-flow Laplace operator defined as:

$$\mathbf{L}_{ij} = \begin{cases} \omega_{ij} = \cot \alpha_{ij} + \cot \beta_{ij} & \text{if } (i,j) \in \mathbf{E} \\ \sum_{(i,k) \in \mathbf{E}}^{k} -\omega_{ik} & \text{if } i = j \\ 0 & \text{otherwise,} \end{cases} \quad (5)$$

and $\alpha_{ij}$ and $\beta_{ij}$ are the opposite angles corresponding to the edge $(i,j)$. Then, we minimize the following quadratic energy:

$$\|\mathbf{W}_C \mathbf{LX}'\|^2 + \sum_i \mathbf{W}_{A,i}^2 \|\mathbf{X}'_i - \mathbf{X}_i\|^2, \quad (6)$$

with diagonal weighting matrices $\mathbf{W}_C$ and $\mathbf{W}_A$, balancing contraction and attraction, the $i$-th diagonal element of $\mathbf{W}_A$ is denoted $\mathbf{W}_{A,i}$ and the coordinate of vertex $i$ is denoted $\mathbf{X}_i$. Subsequently, we update $\mathbf{W}_C^{t+1} = s_L \mathbf{W}_C^t$ and $\mathbf{W}_{A,i}^{t+1} = \mathbf{W}_{A,i}^0 \sqrt{A_i^0/A_i^t}$, where $A_i^t$ and $A_i^0$ are the current and the original one-ring areas, respectively. With the updated vertex positions, a new Laplace operator, $\mathbf{L}^{t+1}$, is computed. We use the following default initial setting: $\mathbf{W}_A^0 = 1.0$ and $\mathbf{W}_C^0 = 10^{-3}\sqrt{A}$, where $A$ is the average face area of the model.

## A.2 Losses and Regularization

### A.2.1 NeRF-Based Scheme Losses

In this section, we explain the losses in the NeRF-based scheme in detail. Additional details are available in BANMo Yang et al. (2022).

$$c_t = \text{MLP}_c(\mathbf{X}_n, v_t, \omega_t)) \tag{7}$$
$$\sigma = \tau(\text{MLP}_{\text{SDF}}(\mathbf{X}_n)) \tag{8}$$
$$\psi = \text{MLP}_\psi(\mathbf{X}_n) \tag{9}$$

Here $\mathbf{x}^t$ means the 2D projection of $\mathbf{X}^t$.

**Reconstruction Loss:**

$$\mathcal{L}_{\text{RGB}} = \sum_{\mathbf{x}^t} \left\| \mathbf{c}(\mathbf{x}^t) - \hat{\mathbf{c}}(\mathbf{x}^t) \right\|_2 \tag{10}$$

$$\mathcal{L}_{\text{silhouette}} = \sum_{\mathbf{x}^t} \left\| \mathbf{s}(\mathbf{x}^t) - \hat{\mathbf{s}}(\mathbf{x}^t) \right\|_2 \tag{11}$$

$$\mathcal{L}_{\text{Optical-Flow}} = \sum_{\mathbf{x}^t, t \to t'} \left\| \mathbf{F}_n^{S,t} - \hat{\mathbf{F}}_n^{S,t} \right\|^2 \tag{12}$$

We compute $\mathcal{L}_{\text{RGB}}$, $\mathcal{L}_{\text{silhouette}}$, $\mathcal{L}_{\text{Optical-Flow}}$ all by minimizing the L2 norm of the difference between the rendered outputs and the ground truth. $\mathbf{c}(\mathbf{x}^t)$, $\mathbf{o}(\mathbf{x}^t)$, $\mathbf{F}_n^{S,t}$ respectively represent the ground truth color, silhouette and optical flow at time $t$ and $\hat{\mathbf{c}}(\mathbf{x}^t)$, $\hat{\mathbf{s}}(\mathbf{x}^t)$, $\hat{\mathbf{F}}_n^{S,t}$ mean the rendered ones at time $t$ correspondingly.

**Feature Loss & Regularization Term**

$$\mathcal{L}_{\text{feature-embedding}} = \sum_{\mathbf{x}^t} \left\| \hat{\mathbf{X}}^*(x^t) - \mathbf{X}^*(x^t) \right\|_2^2 \tag{13}$$

$$\mathcal{L}_{\text{2D-matching}} = \sum_{\mathbf{x}^t} \left\| \Pi^t(\mathcal{W}^{t,\to}(\hat{\mathbf{X}}^*(\mathbf{x}^t))) - \mathbf{x}^t \right\|_2^2 \tag{14}$$

$$\mathcal{L}_{\text{3D-consistency}} = \sum_i \tau_i \left\| \mathcal{W}^{t,\to}(\mathcal{W}^{t,\gets}(\mathbf{X}_i^t)) - \mathbf{X}_i^t \right\|_2^2 \tag{15}$$

$\mathcal{W}^{t,\to}$ and $\mathcal{W}^{t,\gets}$ represent the Blend Skinning and the Backward Blend Skinning involving time $t$ in Sec.3.1. $\tau$ is defined as opacity to give more regularization on the points near the surface. $\hat{\mathbf{X}}^*(\mathbf{x}^t)$ and $\mathbf{X}^*(\mathbf{x}^t)$ are respectively the canonical embeddings in prediction and backward warping by soft argmax descriptor matching Kendall et al. (2017); Luvizon et al. (2019) at time $t$. $\Pi^t$ is the projection matrix at time $t$ from 3D to 2D. For more details refer to BANMo Yang et al. (2022)

| method | AMA-samba | | | AMA-swing | | | Ave. | | |
|---|---|---|---|---|---|---|---|---|---|
| | CD | F@2% | mIoU | CD | F@2% | mIoU | CD | F@2% | mIoU |
| BANMo | 15.3 | 53.1 | 61.2 | 13.8 | 54.8 | 62.4 | 14.6 | 53.9 | 61.8 |
| Ours | 13.1 | 55.4 | 61.8 | 12.7 | 56.2 | 63.2 | 12.9 | 55.8 | 62.5 |

Table 3: Quantitative results on AMA's `swing` and `samba`. 3D Chamfer Distance (cm, ↓), F-score (% ↑), and mIoU (% ↑) are shown averaged over all frames. Note we use half batch size compared with BANMo Yang et al. (2022), which leads to lower accuracy.

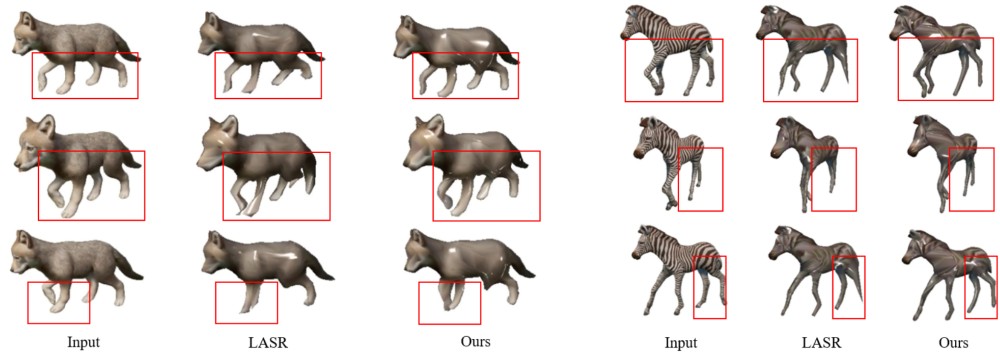

Figure 8: **Rendering Results on PlanetZoo Dataset.** Here we compare our approach with LASR on PlanetZoo's `dog` and `zebra`.

### A.2.2 NEURAL MESH RASTERIZATION-BASED SCHEME LOSSES

We define $\mathbf{S^t}$, $\mathbf{I^t}$, $\mathbf{F}^{2D,t}$ as the silhouette, input image, and optical flow of the input image, and their corresponding rendered counterparts as $\{\tilde{\mathbf{S}}^{\mathbf{t}}, \tilde{\mathbf{I}}^{\mathbf{t}}, \tilde{\mathbf{F}}^{2D,t}\}$

**Reconstruction Loss:** For an NMR-based scheme, we write the total reconstruction loss as a combination of silhouette loss, texture loss, optical flow loss, and perceptual loss (pdist).

$$\mathcal{L}_{\text{reconstruction}} = \beta_1 \left\| \tilde{\mathbf{S}}^{\mathbf{t}} - \mathbf{S}^{\mathbf{t}} \right\|_2^2 + \beta_2 \left\| \tilde{\mathbf{I}}^{\mathbf{t}} - \mathbf{I}^{\mathbf{t}} \right\|_1 + \beta_3 \sigma \left\| (\tilde{\mathbf{F}}^{\mathbf{2D,t}}) - (\mathbf{F}^{2D,t}) \right\|_2^2 + \beta_4 \text{pdist}(\tilde{\mathbf{I}}^{\mathbf{t}} - \mathbf{I}^t) \tag{16}$$

Here, $\beta$'s are the same with LASR, and $\sigma$ is the normalized confidence map for flow.

**Shape Loss:**

To ensure the smoothness of the mesh, Laplacian smoothing is applied Sorkine et al. (2004). The smoothing operation is described per vertex as shown in Equation 17.

$$\mathcal{L}_{\text{shape}} = \left\| \mathbf{X}_i^0 - \frac{1}{|N_i|} \sum_{j \in N_i} \mathbf{X}_j^0 \right\|^2 \tag{17}$$

Where, $\mathbf{X}_i^0$ is coordinates of vertex $i$ in canonical space.

### A.3 IMPLEMENTATION DETAILS

### A.3.1 NERF-BASED SCHEME

Even when frames are available from enough different viewpoints, directly extracting the mesh surface from the NeRF field will result in an inconsistent and non-smooth surface. So closely following BANMo, we utilize the Signed Distance Function (SDF) described in Eq.8. To model dynamic

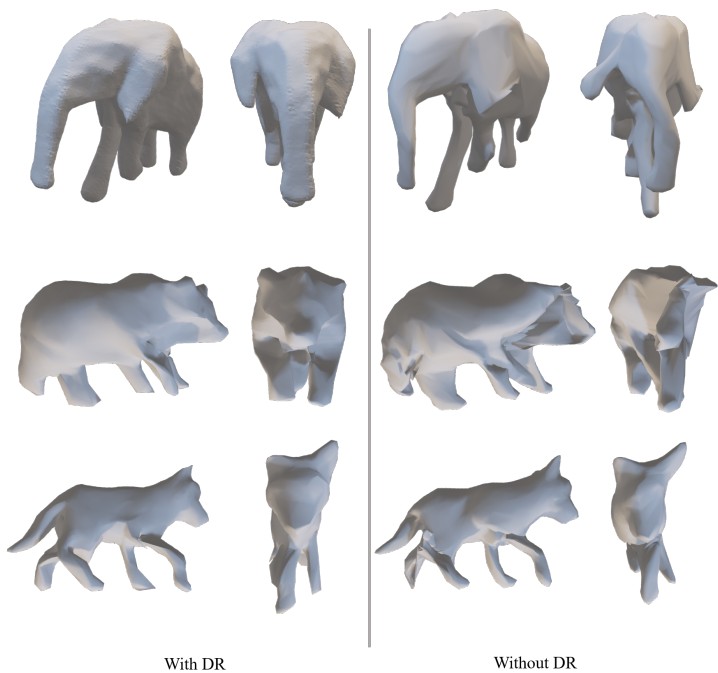

With DR            Without DR

Figure 9: **Reconstruction Outcomes w/ and w/o Dynamic Rigidity.** We present the reconstruction outcomes with and without Dynamic Rigidity on PlanetZoo's `elephant`, `bear`, and `dog` sequences.

scenes, in addition to color $c^t$, view direction $v^t$, and canonical embedding $\psi \in \mathbb{R}^{16}$, time variable should also be included to account for deformations along time $t$. The canonical embedding is designed to adapt to the environmental illumination ($\omega_t$). The MLP$_{\text{SDF}}$ efficiently models the point $\mathbf{X}_n$ as the signed distance to the surface. If the points are outside the surface, the SDF value is negative, and vice versa. $\tau(x)$ is a zero-mean and unimodal distribution accumulating the output from MLP$_{\text{SDF}}$ to transfer the SDF value to density $\sigma$. The zero level-set of the SDF value makes up the extracted surface.

To ensure a fair comparison, our experiments keep most of the optimization and experiment details the same as BANMo Yang et al. (2022). We extract the zero-level set SDF values in the neural radiance field by marching cubes in a predefined $256^3$ grid. Different from BANMo, we initialize the potential rest bones by mesh contraction (Sec.3.1) only one time after the warmup step and allow synergistic iterative optimization (Sec.3.3) to refine the locations and number of bones. AdamW is implemented as the optimizer with 256 image pairs in each batch and 6144 sampled pixels. We train the model on one A100 40GB GPU and empirically find the optimizations stabilize at around the 5th training step with 1h time cost and 15 epochs for each step. The learning rates are set up by a 1-cycle learning rate scheduler starting from the lowest $lr_{\text{init}} = 2 \times 10^{-5}$ to the highest value $lr_{\text{max}} = 5 \times 10^{-4}$ and then falls to the final learning rate $lr_{\text{final}} = 1 \times 10^{-4}$. The rules of Near-far plane calculations and multi-stage optimization follow the same settings as BANMo.

### A.3.2 NEURAL MESH RENDERER-BASED SCHEME

The Neural Mesh Rasterization (NMR)-based approaches directly learn the mesh and potential camera poses rather than extracting them from the neural radiance field of the NeRF-based one. The rest mesh is either provided or initialized by projecting a subdivided icosahedron onto a sphere and further refined (deformed) through a coarse-to-fine learning process. The refinement and deformation are carried out using blend skinning 3.1. In every single forward pass, we employ soft-rasterization to render 2D silhouettes and color images for self-supervised loss computations. This rendering identifies the probabilistic contribution of all the triangles in the mesh to the rendered pixels.

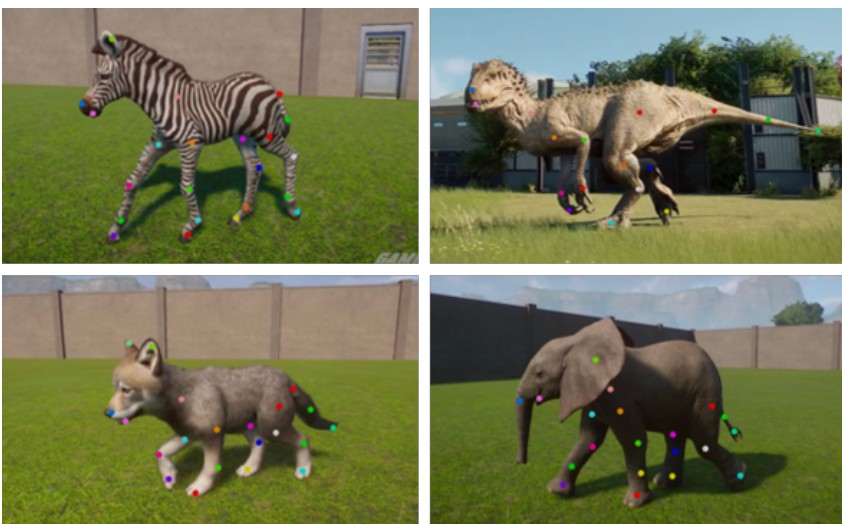

Figure 10: **Annotations for the PlanetZoo Dataset.** Following the annotation guidelines established in BADJA, we have annotated our newly collected PlanetZoo dataset. Consistent with BADJA, annotations are provided every five frames.

We learn the time-varying parameters, including the transformation $\mathbf{T}^t$ and camera parameter $\mathbf{P}_c$, similar to LASR. To generate segmentation masks for the input videos, we used the Segment Anything Model Kirillov et al. (2023). Optical flow was estimated using flow estimators Teed & Deng (2020). Furthermore, we employ the coarse-to-fine refinement approach used by Point2Mesh Hanocka et al. (2020) and LASR Yang et al. (2021a), **but** there is a crucial difference: in the process of refinement, previous approaches fix the number of bones $B$ for each stage and empirically increase the value when moving on to the next stage of the coarse-to-fine refinement. In our case, we start with a large value of $B$ defined from the bones in the initial skeleton. We use the surface flow direction $\mathbf{F}^{S,t}$ to reduce the value of $B$. We iteratively perform this process and end up with the most optimized positions of the bones, which further leads to part decomposition $\mathbf{W}$. For our experiments, we use one A100 40GB GPU. We set the batch size to 4, and epochs are kept the same as LASR.

### A.3.3 MASK AND OPTICAL FLOW GENERATION.

We generate object-level segmentation masks for input videos using Grounding DINO Liu et al. (2023), CSL Zhang et al. (2024) and Segment Anything Model (SAM) Kirillov et al. (2023). By inputting text about the target object in the video, we leverage Grounding DINO to track that object across frames and then we input the detected bounding boxes to SAM/CSL to obtain the segmentation masks. Optical flow was estimated using existing flow estimators Teed & Deng (2020).

### A.4 DATASETS AND METRICS

We demonstrate LIMR's performance on various public datasets including typical articulated objects like humans, reptiles, quadrupeds, etc. To evaluate LIMR and compare our results with previous works, we also adopt two quantitative metrics besides the conventional qualitative visual results like the rendered 3D reconstruction shapes, skeletons, and part assignments. The first one is 2D keypoint transfer accuracy Yang et al. (2021a) meaning the percentage of correct keypoint transfer (PCK-T) Kanazawa et al. (2018); Kulkarni et al. (2020). Given the ground truth 2D keypoint annotations, we label the transferred points whose distance to the corresponding ground truth points is within the threshold as 'Correct'. This threshold is defined as $d_{\text{th}} = 0.2\sqrt{|S|}$. where $|S|$ represents the area of ground truth mask Biggs et al. (2019). The other is the 3D Chamfer distance Yang et al. (2022) which averages the distance between the rendered and ground truth 3D mesh vertices. The matches between each pair of vertices are set up by finding the nearest neighbor.

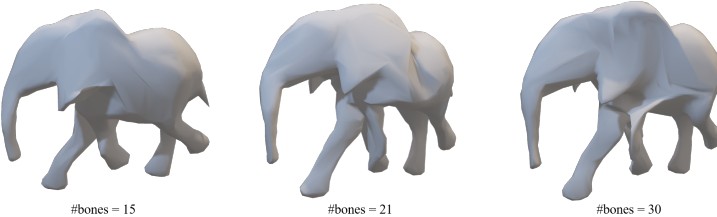

#bones = 15           #bones = 21           #bones = 30

Figure 11: **Reconstruction Outcomes with Different Number of Bones.** We present the reconstruction outcomes with different numbers of bones on PlanetZoo's `elephant` sequences.

**DAVIS & BADJA.** Biggs et al. (2019) derives from DAVIS video segmentation dataset Perazzi et al. (2016) containing nine real articulated animal videos with ground truth 2D keypoints and masks. It includes the typical quadrupeds like dogs, horses, camels, and bears. In addition to the visualization performance, we also adopt the quantitative 2D keypoint transfer accuracy given the ground truth.

**PlanetZoo** is made up of the virtual animal videos collected online. To show our generalization on articulated objects, we collected more quadruped videos and reptile videos with much larger movements compared with the existing public datasets. For example, the animals move much quicker and more complex, and the limbs of dinosaurs with largely different lengths and morphologies are not easy for the existing methods Wu et al. (2022) to use the predesigned per-category ground truth skeletons. We annotated the reasonable ground truth 2D key points using Labelme tools Wada following the real physical morphology across all frames in each collected animal video so that the 2D key points transfer accuracy can be evaluated. The annotation configurations and samples are shown in Figure 10. 2D annotated key points are located on the paw(hand), nose, ear, jaw, knee, tail, neck, elbow, buttock, thigh, and calve, which are common for most animals following morphology. Especially for tails, most tails are long and move flexibly, so the 2D key points are annotated along the tails on the middle point and two ends of the tails.

**AMA human & Casual videos dataset.** Vlasic et al. (2008) records multi-view videos by 8 synchronized cameras. Ignoring the ground truth synchronization and camera parameters, only RGB videos are used in optimization and the experiments only take the decent ground truth 3D meshes in use for calculating the 3D Chamfer distance. The casual videos of `Cat-pikachiu` and `Human-cap` are collected by BANMO Yang et al. (2022) following the same way with AMA datasets but without ground truth meshes.

## A.5 MORE DIAGNOSTICS

**Bone Motion Estimation.** In WIM Noguchi et al. (2022), objects are initially represented as a set of parts, each approximated with an ellipsoid. The method involves learning the pose of each part and then deciding whether to merge two parts based on the relative displacement observed between them throughout the video. In this context, each part is considered a rigid body, having the same SE(3), which allows for the calculation of relative positions by comparing the SE(3) of adjacent parts. However, unlike WIM, where each part is treated as a rigid entity, our approach treats each part as a semi-rigid one. This means that the surface vertices within the same part exhibit similar motions, but with allowances for minor variations.
We implement this concept using blend skinning techniques. The process involves learning the SE(3) of B bones and the skinning weights that describe the relationship between these B bones and N surface vertices. By calculating the linear combination of the SE(3) of bones based on skinning weights, we determine the SE(3) for each vertex. This approach allows for a more flexible description of each vertex's motion.
Intuitively, we aim to correlate each part with a single bone, but this one-to-one correspondence often does not hold true. In practice, we find that skinning weights are typically non-sparse, with the motion of surface vertices being determined by multiple bones. Consequently, the motion of each semi-rigid part is usually influenced by several nearby bones, especially during the initial stages after skeleton initialization when the number of bones is large. In such cases, representing the motion of one part with a single bone becomes unreasonable, a fact supported by our initial attempts.
To address this, we consider using 2D optical flow to determine the 2D motion of surface vertices

and skinning weights to calculate the 2D motion of each part. This process helps in deciding whether adjacent parts or bones should be merged. Here, 'bone' refers to the one with the highest weight in relation to the part, but it might not be the only bone affecting the part's motion. Our experiments demonstrate that this method yields satisfactory results.

**NeRF-based methods VS NMR-based methods** NeRF-based methods such as BANMo have showcased impressive reconstructions. These approaches involve the learning of a neural radiance field, from which a mesh is extracted. However, learning such a field requires a substantial amount of data, typically including multiple videos and viewpoints, which can be challenging to acquire. These approaches experience a significant drop in performance when provided with limited data and views as shown in Fig.3. On the other hand, works based on neural mesh renderer directly train models to generate a mesh, yielding satisfactory results in limited data settings, such as a single monocular video with fewer than a hundred input frames. We demonstrate the applicability of our approach in both of these scenarios and showcase significant qualitative and quantitative performance improvements in cases with limited data.

**Failure Cases due to Wrong Camera Predictions.** BANMo exhibits high sensitivity to extensive camera motion within videos. This is attributed to its reliance on PoseNet to estimate the camera pose from the initial frame. Significant pose variations in subsequent frames can severely degrade its performance. For instance, as shown in Fig.3 in the `zebra` experiment, it learns two heads incorrectly. As illustrated in Fig.13, the Symmetry Loss is adversely affected by videos exhibiting long-range camera motions. Incorrect camera parameter predictions hinder the algorithm's ability to deduce a plausible symmetry plane. Fig.13 displays various views of the reconstructed `zebra`, highlighting the inaccuracies in the symmetry plane. Consequently, we opted to exclude the symmetry constraints.

## A.6 DIFFERENCE BETWEEN LIMR AND EXISTING METHODS

As shown in Tab. 2, we list the comparison among LIMR and existing methods regarding different settings. LIMR tackles more challenging but realistic problems which are: 1) In the wild, there are rarely ground truth (GT) 3D shapes, skeleton templates, and camera poses provided. 2) We cannot ensure multiple videos of moving objects containing different actions and views can be provided. Works Kuai et al. (2023); Yang et al. (2023); Wu et al. (2023a) like CASA Wu et al. (2022), and SMAL Zuffi et al. (2017) requiring instance-specified 3D shapes/skeleton/both templates of each object for motion modeling will be undoubtedly limited by their generalization to other Out-Of-Distribution objects without GT 3D information as input. Some approaches such as WIM Noguchi et al. (2022), CAMM Kuai et al. (2023), RAC Yang et al. (2023), and BANMo Yang et al. (2022) require providing accurate camera poses and multiple videos (more than 1000 frames) with diverse views in order to provide decent results. Methods like LASR Yang et al. (2021a) and VISER Yang et al. (2021b) do not require a pre-built template as input. However, due to the absence of a skeleton that can provide structural information about the object, they often fail to achieve optimal results. In contrast, LIMR used the learned near-physical skeleton to facilitate modeling the motions of moving articulated objects instead of using virtual bones in LASR Yang et al. (2021a), BANMo Yang et al. (2022), and ViSER Yang et al. (2021b), while significantly minimizing the requirements for input.

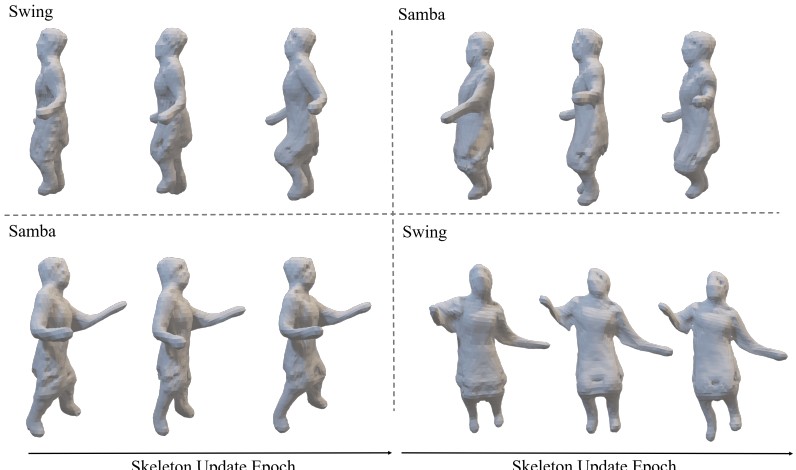

Figure 12: **Reconstruction Outcomes during Skeleton Updates.** We present the reconstruction outcomes at various epochs post-skeleton update for AMA's `swing` and `samba` sequences.

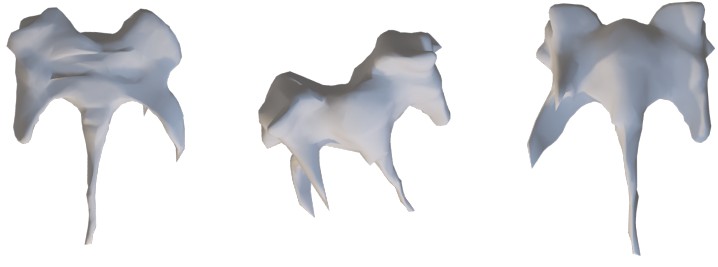

Figure 13: **Failure Cases due to Symmetry Loss.**

Table 4: Table of Notations

| Symbol | Description | Dimension |
|---|---|---|
| $B$ | Number of Bones | – |
| $E$ | Number of Edges in Surface Mesh | – |
| $J$ | Number of Joints | – |
| **Representations Notations** | | |
| $\mathcal{R}_e$ | Explicit Representations | – |
| $\mathcal{R}_i$ | implicit Representations | – |
| **Representation Parameters** | | |
| $\mathbf{M}$ | Canonical Surface Mesh and Color | – |
| $\mathbf{P}_c^t$ | Camera Parameters | – |
| $\mathbf{W}$ | Skinning Weights | $\mathbf{W} \in \mathbb{R}^{N \times B}$ |
| $\mathbf{R}$ | Rigidity Coefficient | $\mathbf{R} \in \mathbb{R}^E$ |
| $\mathbf{T}^t$ | Time Varying Transformation | $\mathbf{T} \in SE(3)$ |
| **Skeleton Notations** | | |
| $\mathbf{S_T}$ | Skeleton | – |
| $\mathbf{B}$ | Bones | $\mathbf{B} \in \mathbb{R}^{B \times 13}$ |
| $\mathbf{J}$ | Joints | $\mathbf{J} \in \mathbb{R}^{J \times 5}$ |
| **Mest Contraction** | | |
| $\mathbf{X}$ | Vertices Coordinates | – |
| $\mathbf{E}$ | Edges between Vertices | – |
| **Properties of 3D Points** | | |
| $\mathbf{c}$ | Color of a 3D point | $\mathbf{c} \in \mathbb{R}^3$ |
| $\sigma$ | Density of a 3D point | $\sigma \in \mathbb{R}$ |
| $\psi$ | Canonical embedding of a 3D point | $\psi \in \mathbb{R}^{16}$ |
| **Bone Components** | | |
| $\mathbf{C}/\mathbf{B}[:,:3]$ | Gaussian Centers Coordinates | $\mathbf{C} \in \mathbb{R}^{B \times 3}$ |
| $\mathbf{Q}/\mathbf{B}[:,3:12]$ | Precision Matrix | $\mathbf{Q} \in \mathbb{R}^{B \times 9}$ |
| $\mathbf{L}/\mathbf{B}[:,12]$ | Bone Length | $\mathbf{L} \in \mathbb{R}^B$ |
| **Joint Components** | | |
| $\mathbf{J}[:,:2]$ | Index of Two connected Bones | $\mathbb{R}^{J \times 2}$ |
| $\mathbf{J}[:,2:]$ | Joint Coordinates | $\mathbb{R}^{J \times 3}$ |
| **Optical Flow Notations (at time $t$)** | | |
| $\mathbf{F}^{2D,t}$ | 2D Optical Flow | $\mathbf{F}^{2D,t} \in \mathbb{R}^{H \times W}$ |
| $\mathbf{F}^{B,t}$ | 2D Bone Motion Direction | $\mathbf{F}^{B,t} \in \mathbb{R}^{B \times 2}$ |
| $\mathbf{F}^{S,t}$ | Surface Flow Direction | $\mathbf{F}^{S,t} \in \mathbb{R}^{S \times 2}$ |
| $\mathcal{V}^t$ | Visibility Matrix | $\mathcal{V}^t \in \mathbb{R}^{N \times 1}$ |
| **Blend Skinning Functions** | | |
| $\mathcal{W}^{t,\rightarrow}(\mathbf{X}^0)$ | Forward Blend Skinning from $\mathbf{X}^0$ to $\mathbf{X}^t$ | - |
| $\mathcal{W}^{t,\leftarrow}(\mathbf{X}^t)$ | Backward Blend Skinning from $\mathbf{X}^t$ to $\mathbf{X}^t$ | - |
| **2D Notations** | | |
| $\mathbf{S}/\tilde{\mathbf{S}}$ | Input silhouette/ Observed silhouette | $\mathbf{S}/\tilde{\mathbf{S}} \in \mathbb{R}^{H \times W}$ |
| $\mathbf{I}/\tilde{\mathbf{I}}$ | Input Image / Observed Image | $\mathbf{I}/\tilde{\mathbf{I}} \in \mathbb{R}^{H \times W}$ |
| $\mathbf{F}^{2D}/\tilde{\mathbf{F}}^{2D}$ | 2D Input OF / 2D observed OF | $\mathbf{F}^{2D}/\tilde{\mathbf{F}}^{2D} \in \mathbb{R}^{H \times W}$ |

