# OpenReview forum: "Learning Implicit Representation for Reconstructing Articulated Objects"
_ICLR.cc/2024/Conference — ICLR 2024 poster_

### Official Review · Reviewer_Wx9e · 2023-10-28

**Soundness:** 3 good
**Presentation:** 3 good
**Contribution:** 3 good
**Rating:** 5
**Confidence:** 3

**Summary:**

Input:  One or more monocular video containing moving object that exhibits articulations;
Output: 3D shape of moving object in the input video

The paper presents an iterative synchronization framework that predicts the 3D shape of moving objects, specifically, living beings such as animals and humans, from input monocular videos. The 3D shape is decomposed into two parts – explicit (the surface and its color), and implicit (the skeleton, the object’s semi-rigid parts, their motions, and the articulated part motion parameters given by rigidity coefficients).
It presents a Synergistic Iterative Optimization scheme for Shapes and Skeletons, henceforth referred to as SIOS^2, that learns both explicit and implicit representations (as defined above), iteratively. This is analogous to the Expectation Maximization (EM) algorithm in Machine Learning.

During the E phase, the implicit skeleton is improved based on the current explicit rep. In this step, the 2D motion direction of each semi-rigid part, i.e., the bone, is calculated and distances between connected joints at the end of the bone are measured. The skeleton is then updated using the consistency of direction of the optical flow within each semi-rigid part, and the constancy (not consistency) of distances between connected joints, i.e., bone length, across the video frames.

With this E step, the 3D shape is updated in the M step using the updated skeleton (obtained from the E step).



Dataset used:

The paper uses the following different kinds of datasets that contain videos of articulated living beings.
DAVIS – Animals
PlanetZoo – Animals
AMA – Humans
BANMO - Humans



Underlying Neural Network:

For the so-called “implicit representation learning”, i.e., skeleton-associated representation learning, I do not see any neural network being used. Pls confirm if this is this is the case. What I understand from the paper is that this stage builds upon existing techniques and uses non-neural network-based optimization to obtain Bones, Vertices and Joints.

The neural part is during the learning of explicit representation, which learns the surface of the articulated living being. This is where two different kinds of neural networks are used.
The first one is a NeRF model with MLPs on Signed Distance Fields (SDFs). The second model is a Neural Mesh Rasterizer (NMR) model that is used to enforce consistency in the 2D domain.


Loss functions:

Similar to Volumetric-based rendering methods for 3D surface reconstruction (Yariv et al. 2021), the NeRF model uses the following loss terms:
L_NeRF = L_recon + L_feature + L_3DConsistency, where
L_recon = L_silhouetter + L_rgb + L_opticalflow
L_feature is the 3D feature embedding loss that minimizes the distance between the canonical embeddings of the 3D shape from prediction and backward warping, and
L_3Dconsistency  is used to ensure forward-deformed 3D points match their original location upon backward-deformation.

L_NMR = L_recon&perceptual + (alpha x L_shape) + (lamda x L_DR), where
L_recon&perceptual = is the same as L_NeRF but with additional perceptual loss,
L_DR is the dynamic rigid loss and L_shape is the shape regularizer



Quantitative Metric:

Table 1- 2D Keypoint Transfer Accuracy,
Table 3- Chamfer Distance on shape reconstruction task from multiple videos

**Strengths:**

1) The paper addresses a challenging task of estimating deformations on 3D shapes of humans and animals, in terms of both surface/skin and skeleton articulations, from one or more monocular videos. In doing so, no ground truth 3D supervision is used.

2) The clarity of writing is reasonable, something that does not appear to be difficult to follow for folks working in this area. And most of the related works in the context of this paper are discussed to the degree that is possible within the page limit.

3) The limitations are aptly discussed, which educate the readers about the gaps to be filled despite the attempts/contributions in the paper.

4) The SIOS2 algorithm, which is an interleaving-iterative algorithm, is intuitive and showing that this works on this task (i.e., the task of estimating 3D shapes of living beings that undergo articulations, from video inputs) adds value to such co-analysis based techniques in Visual Computing research.

**Weaknesses:**

1) The presented pipeline is overly complicated, with the iterative SIOS2 algorithm coupled with optical flow estimation to constrain the prediction of shape skeleton based on consistency across different frames of the video. And so, the method may not be easily extendable to potential new projects in this direction. Plus, I have concerns about the reproducibility of the method.

2) Another major concern I have is the lack of diversity of shape samples used to demonstrate the superiority of the reconstruction/rendering results from LIMR (the paper’s approach). Specifically, Fig 3 shows limited diversity of shape samples (a Camel and a Horse). And I have not seen much diversity in the Appendix either (Fox, Zebra and Elephant; I would consider a Zebra equivalent to a Horse). Is this because PlanetZoo and DAVIS datasets only have these classes of animals?

3) The rendering results are poor in quality, see Fig 3 and 7. It is difficult to evaluate the goodness of the results from these visualizations. It will help the paper if the renderings are improved.

4) The discussion in Section 4.3 are good. A similar discussion will be good for “Quantitative Comparison” paragraph that compares LIMR with LASR and ViSER. This is missing and the “Quantitative Comparison” paragraph is simply a plain run of Table1. Some intuition on the performance comparison is missing, perhaps due to the page limit?

5) Having just CD as the eval metric for reconstruction task does not provide a full picture. Additional evaluation metrics such as EMD, F-Score, Normal Consistency and even IoU will provide a good picture of the quality of reconstructions.

**Questions:**

1) I am not sure if the skeleton representation should be termed an implicit representation. It is intrinsic/internal to the shape. In today’s context of neural fields, using “implicit rep” to refer to skeletonization is misleading.

2) The minimum number of videos that can be input to the method is 1. What is the max?

3) And is there a limit on the video length? What is the minimum number of frames in the input video that should contain the articulating living being for LIMR to work? Can it work if there are just two frames that contain the living being? Just one frame is not possible I believe?

4) What is the resolution of the video input?

5) What is the resolution of video frames processed by the LIMR framework? Same as video input or do you downsize?

6) Can LIMR work with grayscale videos?

7) What is the range of number of bones and bone joints (max and min) that this method can work on?

8) Can videos containing multiple objects be used as input? If so, can all the objects be recovered in terms of their shapes? If not, which object will be selected in such a case? I guess the method will need an additional pre-processing step to detect object classes and/or add a saliency module. Can you comment on these? These are all important considerations that need to be discussed in the limitations section.

---

> ### Author Response · Authors · 2023-11-17
> **Respond to Reviewer Wx9e -- Part 1(a)**
>
> We are thankful to Wx9e for noting that (1) our paper addresses a challenging task, (2) the clarity of writing is reasonable, (3) most of the related works are discussed, (4) the limitations of our work are aptly discussed, and (5) our SIOS2 algorithm is intuitive, works and adds value to literature.
>
> We greatly appreciate Wx9e for the weaknesses noted and the questions they have raised. We outline below our response and how we plan to use these comments to improve the quality of our paper.
>
> 1a. Complexity: As shown in Alg.1 from the Appendix, LIMR iteratively updates the reconstruction model (shape, time-varying parameters) and the skeleton, and the skeleton is updated according to the 2 intuitive physical constraints. We will try to make LIMR more easy to follow and not seem over-complicated to the readers.
>
> 1b. Extensibility: Regarding extensibility to new projects, we demonstrate LIMR's scalability to NMR-based methods (e.g.,[5]) and NeRF-based methods (e.g.,[3]), showing it can adapt to new tasks that use blend skinning for object pose manipulation with video input. Our method eliminates the need for ground truth skeletons required by existing approaches and does not require any additional annotations. Moreover, LIMR is able to obtain a skeleton comparable to that obtained by RigNet [8], which is trained with massive 3D data (mesh, skeleton, and skinning weights). We will include comparisons in the paper and examples are here https://drive.google.com/file/d/18evWJ-uBl7U7NoHH0PN1tKiz_7FAFX1U/view for immediate reference. Therefore, LIMR can be easily extended to methods like [2] that leverage RigNet [8] for the skeleton.
>
> 1c. Reproducibility: To ensure reproducibility, we have provided all implementation details of LIMR in both NMR-based and NeRF-based contexts in the Appendix (Section A.3). Additionally, we have described each loss function utilized in our study in detail (Section A.2). Following publication of our paper, we will immediately release the code for public access and reference.
>
> 2. Diversity of Shape Samples: The diversity of the PlanetZoo and DAVIS datasets is limited. The structural, appearance, and size differences captured are indicated by the differences among, e.g., horse, dinosaur, giraffe, elephant, and tiger. We will include additional results for more animals, e.g., bear, cow, giraffe, dinosaur and tiger in the paper.
>
> 3. Rendering Image Quality: We will surely provide higher-quality renderings. In the meantime, we believe that Figures 3 and 7 already illustrate the improvements made by LIMR, particularly in challenging image regions within the red boxes. We will show video results on the project pages; meanwhile, we have added some examples at https://drive.google.com/drive/folders/1IUrSi-1VcBtr4FbIRrRSt9PuVrKYFE14 for immediate reference.
>
> 4. More Discussion about Quantitative Performance: We agree that an additional, Sec 4.3-like discussion about quantitative comparisons will be good to include, but we had to limit these details in Section 4.2 due to space constraints. We will add this to the Appendix.
>
> 5. Additional Evaluation Metrics: We will add evaluation metrics of F-score and IoU. F-score clearly brings out our improvements over BANMo. Our IoU results are less pronounced due to the dominance of the silhouette loss component within reconstruction loss (In certain instances, LASR [5] may incorrectly deform the shape, but still yield a favorable IoU score. For example, rather than positioning the legs correctly, it erroneously stretches the area beneath the belly without correspondingly hurting the IoU score.

---

> ### Author Response · Authors · 2023-11-17
> **Respond to Reviewer Wx9e -- Part 1(b)**
>
> Responses to Questions:
>
> 1. Terminology: We agree and will replace our term 'implicit' with 'latent'.
>
> 2. Video Length: There's no maximum limit to video number and length. LIMR can produce satisfactory results with approximately 15 frames, but we can assure accurate reconstruction for only the viewed angles. We have not tested performance with fewer frames as they may not fully contain an action. Exploring the minimal frame limit is a potential avenue for future work.
>
> 3. Video Resolution: For NMR-based LIMR, the training image resolution is 1920x1080 same as in [5]. For NeRF-based LIMR, images are downsampled to 64x64, matching the settings used in [3]. We will add these details to the paper.
>
> 4. Grayscale Videos: Although not included in the paper, we had performed experiments showing that removing texture (color) related losses has a negligible effect on the resulting mesh quality (likely because texture information is already minimal for some animals which are of almost a single color, e.g., camels). This suggests that LIMR will be negligibly affected if we use gray-scale videos instead of color videos.
>
> 5. Bone Number Range: Initially, we worked with about 150-200 bones; later we could reduce this number to around 10 (e.g., for a tiger) while maintaining good performance.
>
> 6. Multiple Objects: We use GroundingDINO [6] and SAM [7] to obtain a mask for the target object. When multiple objects are present, GroundDINO and SAM can track and segment the target object using textual prompts.  LIMR reconstructs the object corresponding to the mask. Thus, LIMR works for one object at a time, selected by the user, from among the multiple objects detected and tracked. We will add these details and provide code in the supplementary materials or on GitHub.
>
>
> [1] Noguchi, Atsuhiro, et al. "Watch it move: Unsupervised discovery of 3D joints for re-posing of articulated objects." Proceedings of the IEEE/CVF Conference on Computer Vision and Pattern Recognition. 2022.
>
> [2] Kuai, Tianshu, et al. "CAMM: Building Category-Agnostic and Animatable 3D Models from Monocular Videos." Proceedings of the IEEE/CVF Conference on Computer Vision and Pattern Recognition. 2023.
>
> [3] Yang G, Vo M, Neverova N, et al. “Banmo: Building animatable 3d neural models from many casual videos” Proceedings of the IEEE/CVF Conference on Computer Vision and Pattern Recognition. 2022.
>
> [4] Wu, Shangzhe, et al. "Magicpony: Learning articulated 3d animals in the wild." Proceedings of the IEEE/CVF Conference on Computer Vision and Pattern Recognition. 2023.
>
> [5] Yang, Gengshan, et al. "Lasr: Learning articulated shape reconstruction from a monocular video." Proceedings of the IEEE/CVF Conference on Computer Vision and Pattern Recognition. 2021.
>
> [6] Liu, Shilong, et al. "Grounding dino: Marrying dino with grounded pre-training for open-set object detection." arXiv preprint arXiv:2303.05499 (2023).
>
> [7] Kirillov, Alexander, et al. "Segment anything." arXiv preprint arXiv:2304.02643 (2023).
>
> [8] Xu, Zhan, et al. "Rignet: Neural rigging for articulated characters." arXiv preprint arXiv:2005.00559 (2020).

---

> > ### Comment · Reviewer_Wx9e · 2023-11-17
> > **Rebuttal Acknowledgment**
> >
> > Authors,
> >
> > Thanks for the response. I acknowledge that I have read through the rebuttal.
> >
> > I am not convinced about the extendability of the method, given the complicated workflow. In addition, as Reviewer dR3E too pointed out, it is difficult to understand which component in the framework is important toward the end goal.
> >
> > In addition, given the quality and diversity of results, and also looking at other reviews, I am not inclined to change my score.

---

> > > ### Author Response · Authors · 2023-11-17
> > > **Respond to Reviewer Wx9e -- Part 2**
> > >
> > > (A) Importances of Components: To the best of our understanding and experience, we feel that in a challenging task like the one we have attempted in this paper (single, monocular, short video without prior template), it is not easy to depend on one or a few sources of information to estimate the shape, joint structure, and articulated motion. Because of the highly compressed, convoluted, and time-varying interdependence of the sources, we think any estimation algorithm will have to make joint use of one or more of the information sources captured in our formulation and loss function, their relative contributions possibly varying from time to time. We therefore think that there is no single component in the framework that is consistently important to reach the end goal. This distributed nature of information is confirmed by the related previous papers [1,2,3,4,5], which also do not quantify the relative contributions of the different sources beyond presenting the ablations.  In our paper Sec. 4.3, we have presented the following ablation studies to demonstrate the effectiveness and analyze the reasons for the effectiveness of the various modules:
> > > 1. Physical Skeleton (ours) vs Virtual Bones
> > > 2. Refined Skeleton (ours) vs Initial Skeleton (Mesh Contraction)
> > > 3. Different Thresholds for Skeleton Refinement
> > > 4. Impact of Video Content on Skeleton
> > > 5. Efficacy of Dynamic Rigidity and Part Refinement
> > >
> > > Although Reviewers Z1Zw and gF7P have expressed satisfaction with the range of these ablations, we would be grateful if you (Wx9e) could help us with which other ablations we could add; or how else could we specify module importances (given that there is no single consistently dominant module). We look forward to your suggestions about any additional experiments, analyses, etc. that we could include to improve our paper.
> > > In addition to 1-5 above, we also plan to provide ablation studies on different parameters of the mesh contraction module.
> > >
> > > (B) Extendability: As mentioned in the paper and the rebuttal, we have demonstrated successful extensions of our method to the rather different settings of the NMR-based methods (such as LASR) and NeRF-based methods (e.g., BANMo).
> > > These extensions show that our method can effectively learn latent structures and hence has the ability to extend to methods. We seek your input on what else we could do to further demonstrate extensibility, and we will be glad to include such experiments/results/analyses.
> > >
> > > (C) Quality and Diversity of Results: To enable comparison with baselines of related previous work [2,3,4,5], we have obtained our results for input videos with the same diversity (variety of animals). We seek your input about any specific types of input videos (e.g., of any other animals, such as giraffe, dinosaur, tiger, or other objects?) that would better demonstrate the diversity of our results. We will be glad to improve the quality of our paper by adding these results.
> > >
> > > [1] Noguchi, Atsuhiro, et al. "Watch it move: Unsupervised discovery of 3D joints for re-posing of articulated objects." Proceedings of the IEEE/CVF Conference on Computer Vision and Pattern Recognition. 2022.
> > >
> > > [2] Kuai, Tianshu, et al. "CAMM: Building Category-Agnostic and Animatable 3D Models from Monocular Videos." Proceedings of the IEEE/CVF Conference on Computer Vision and Pattern Recognition. 2023.
> > >
> > > [3] Yang G, Vo M, Neverova N, et al. “Banmo: Building animatable 3d neural models from many casual videos” Proceedings of the IEEE/CVF Conference on Computer Vision and Pattern Recognition. 2022.
> > >
> > > [4] Wu, Shangzhe, et al. "Magicpony: Learning articulated 3d animals in the wild." Proceedings of the IEEE/CVF Conference on Computer Vision and Pattern Recognition. 2023.
> > >
> > > [5] Yang, Gengshan, et al. "Lasr: Learning articulated shape reconstruction from a monocular video." Proceedings of the IEEE/CVF Conference on Computer Vision and Pattern Recognition. 2021.

---

> > > ### Author Response · Authors · 2023-11-22
> > > **Respond to Reviewer Wx9e -- Part 3**
> > >
> > > Thanks again for your efforts and suggestions for this paper. The deadline for the author-reviewer discussion is approaching. After discussing with reviewers, updating the paper, and supplementing the experiments, most of the issues have been addressed. We are also very keen to know whether your concerns (some of which are shared with other reviewers) have been resolved. Do you have any other questions? We are more than willing to resolve any of your doubts through our analysis, discussion, or further experiments.

---

> > > > ### Author Response · Authors · 2023-11-23
> > > > **Respond to Reviewer Wx9e -- Part 4**
> > > >
> > > > As we have mentioned in Part 3 of our (Nov 22) response above, we have responded to your three main comments of Nov 17, namely: (1) Extendability, (2) Relative values (ablations) of various components of our method, and (3) Quality and Diversity of our results. We have also updated our manuscript to more clearly explain these three aspects. We hope that these clarifications and updates have addressed your concerns.
> > > >
> > > > We are eager to know if we have indeed been able to answer your questions. If not, could you please let us know about any remaining issues so we can address them? Thanks.

---

### Official Review · Reviewer_2K9U · 2023-10-29

**Soundness:** 3 good
**Presentation:** 2 fair
**Contribution:** 2 fair
**Rating:** 6
**Confidence:** 5

**Summary:**

The paper tackles the problem of articulate 3D shape reconstruction from a single video, similar to the setup of LASR. The main contribution is a method for automatic skeleton structure discovery. They also introduced a variant of local rigidity loss that accounts for the flexibility of surface points.

The method is evaluated on DAVIS and Planet Zoo datasets, out-performing LASR/ViSER. It also shows qualitative results on AMA and  BANMo's videos, with better performance than BANMo.

**Strengths:**

**Originality**

The paper introduces a few new techniques I found interesting.
- The Dynamic Rigid (DR) loss aims to encourage local rigidity (similar to ARAP) in a more flexible way. Instead of applying ARAP uniformly on the surface, it is weighed by the "Rigidity coefficient", which is a function of skinning weights. In this way, edges with peaky skinning weight will receive more penalty as they should move more rigidly. The effect is validated in Tab 1.
- They leverage 2D motion (flow) constraints for skeleton discovery. To achieve this, an inverse blend skinning method is used to aggregate a 2D flow map to bones. Bone with similar 2D flow will be merged.

**Weaknesses:**

**Presentation**
- In general, I feel the presentation could be significantly improved and the paper could be made much stronger.
- The usage of certains terms made it difficult to follow. For example:
  - "semi-rigid part assignment" can be replaced by "skinning weights", which is a term that already exists in graphics and vision literature
  - "implicit motion representation": I would think it is explicit representation, given they are actually a set of rigid transformations that are explicitly defined. I think "Internal / latent" representation better describes skeleton movements.
  - "inverse blend skinning" (IBS) is an unconventional term and needs more explanation and highlighting. Indeed, one may find naming it inverse of blending skinning not accurate. Blend skinning maps a set of points X, rigid transformations G, and skinning weights W to another set of points X'=WGX. The inverse could be finding G and W from X'. My understanding is that the paper uses IBS to map vertex properties (e.g. 2D flow) to bones with an existing W. Is this correct?
- The method section could be better structured with some "pointer" sentences. For example, bone motion direction is introduced in Eq (4) but it was not mentioned why we need to compute them. It only appears in Sec 3.3. With this, I would suggest either move IBS to Sec 3.3, or pointing the reader to Sec 3.3 when motivating Eq (4).
- It is not immediately clear what purpose Sec 3.2 aims to serve. The rest of the paper describes the shape representation as meshes, but this section talks about the pros and cons of mesh vs neural fields. If the goal is to how that method can be applied to neural fields as well, I feel this can be moved to implementation details.

**Related works**
- Some related works are not discussed.
  - [A]-[B] find skeleton from video
  - [C]-[D] find skeleton from 3D point/mesh sequences
  - [E] finds skeleton from image features
- Since [A]-[D] also deal with motion data, some method-level discussion or even comparison is needed. I think [A] is particularly related as they also search for the skeleton structure and optimize shape in an alternating fashion.


Experiments
- To strengthen the experiments, the bone discovery results could be separately evaluated and compared against some existing work, such as Rignet or Morig[D].
- For bone merging, one alternative is to use the similarity of SE(3) of bones. The effect of "2D flow" vs "3D rigid transform" could be ablated through an experiment.


[A] Noguchi, Atsuhiro, et al. "Watch it move: Unsupervised discovery of 3D joints for re-posing of articulated objects." Proceedings of the IEEE/CVF Conference on Computer Vision and Pattern Recognition. 2022.

[B] Kuai, Tianshu, et al. "CAMM: Building Category-Agnostic and Animatable 3D Models from Monocular Videos." Proceedings of the IEEE/CVF Conference on Computer Vision and Pattern Recognition. 2023.

[C] Le, Binh Huy, and Zhigang Deng. "Robust and accurate skeletal rigging from mesh sequences." ACM Transactions on Graphics (TOG) 33.4 (2014): 1-10.

[D] Xu, Zhan, et al. "Morig: Motion-aware rigging of character meshes from point clouds." SIGGRAPH Asia 2022 Conference Papers. 2022.

[E] Yao, Chun-Han, et al. "Lassie: Learning articulated shapes from sparse image ensemble via 3d part discovery." Advances in Neural Information Processing Systems 35 (2022): 15296-15308.

**Questions:**

1. Fig 1 caption: Is the rigidity coefficient a parameter? I thought R could be directly computed from D.
2. It would be more convincing to include video results and comparisons.
3. BANMo's result in Fig 3 appears particularly bad. Is there any explanation? The results on Swing also appear coarse. What is the resolution of the marching cubes? Also, I overlaying the skeleton with the mesh similar to Fig 4. would make the results more compelling.

Other comments
1. Fig 6 is nice as it shows the initial configuration of the skeleton. It would be nicer to show the progression of the skeleton over iterations.

---

> ### Author Response · Authors · 2023-11-17
> **Respond to Reviewer 2K9U -- Part 1(a)**
>
> We thank 2K9U for noting the (1) originality of and flexibility of our loss function DR, (2) its validation in the paper, and (3) our use of inverse blend skinning for skeleton discovery.
>
> We greatly appreciate 2K9U for the weaknesses noted and the questions they have raised. We outline below our responses and how we plan to use these comments to improve the quality of our paper.
>
> Presentation Improvements:
>
> We thank 2K9U for valuable suggestions about improving the readability, and we will implement these suggestions as explained below.
>
> 1. We agree with the suggestions to rename some of our terms for better understanding, as follows.
>
> (a) We will replace our term "semi-rigid part decomposition" with "skinning weights," which is a more conventional term within the graphics and vision literature.
>
> (b) We will replace our term "implicit motion representation" with "latent representation," to avoid confusion with explicit transformations; we agree that "latent" better captures the intended underlying skeletal movements that are not directly observable.
>
> (c) We will replace our term "inverse blend skinning" (IBS) with simply "blend skinning" to align with standard practices and better convey its true role; we will also provide a more detailed explanation of it in mapping vertex properties to bones with an existing set of skinning weights.
>
> 2. We will add pointer sentences to guide the reader through our methodology section. For instance, we will clearly state and explain the relevance, with reference to Section 3, of the rationale behind computing bone motion direction introduced in Eq (4).
>
> 3. Purpose of Section 3.2: We agree that the discussion in this section (about our method's ability to integrate with NeRF-based and NMR-based approaches) may distract the reader from the primary focus of our method, and will be better located as part of implementation details; accordingly, we will move this discussion to the implementation section.
>
> Related Works and Discussion:
>
> (a) We thank 2K9U for pointing us to additional related works. We find these useful, will refer to them, and contrast our approach with these methods, particularly highlighting the differences in requirements for prior information and applicability in in-the-wild scenarios.
>
> (b) We agree that a comparison of our work with the approaches mentioned by 2K9U, particularly with Rignet [8] will strengthen the paper. We will include these in our discussion and also note the differences in that they require extensive ground truth 3D annotations such as high-quality mesh, skeleton, and skinning weights, which are expensive and challenging to obtain. An example of a comparison between RigNet and LIMR can be found at https://drive.google.com/file/d/18evWJ-uBl7U7NoHH0PN1tKiz_7FAFX1U/view?usp=sharing. These results show that our LIMR yields skeletons comparable to those given by RigNet, even though we do not require any 3D supervision.
>
> (d) Regarding alternative bone merging strategies: We have explored the use of the similarity of SE(3) transformations for bone merging. However, we found this approach to be not ideal due to the non-sparsity of skinning weights, which makes the motion of vertices (parts) depend on multiple bones, especially in the early stages with around 200 bones. Therefore, representation of the motion of each part by the SE(3) of one bone will lead to unreliable motion similarity assessments. Upon further exploration, we discovered that applying optical flow warp techniques significantly enhances the accuracy and logical coherence of part motions. We will add this discussion in the updated paper.
>
> (e) General Discussion on Our Method vs Related Works: Methods like [2], [6], and [10] require a ground truth skeleton or pre-trained model like Rignet to obtain the skeleton. [1] gives decent results given multiple videos captured from different views and ground truth camera poses. In contrast, as discussed in sections  A.5 (NeRF-based methods VS NMR-based methods), such methods ([1], [2], [3], [4], [6]) struggle to obtain decent results given a short monocular video. Among current methods, Hi-Lassie [7] shows the best results in our setting (short monocular video without extra annotations and camera pose), and we will add a comparison with it in our paper; illustrative results we have obtained are at the link:  https://drive.google.com/file/d/1VRdZeV0GK3pELwL8Y7u8OYHuXW9xWAoI/view
>
> In contrast, we believe that our method's ability to learn a satisfactory skeleton structure from a short monocular video without any prior skeletal, shape information, or camera poses is a significant contribution. We will revise our paper to emphasize this distinction, include the limitations of other methods in in-the-wild scenarios, and provide a visual comparison.

---

> ### Author Response · Authors · 2023-11-17
> **Respond to Reviewer 2K9U -- Part 1(b)**
>
> Responses to Specific Questions:
>
> (a) Fig 1 Caption: The rigidity coefficient is indeed derived from skinning weights; we will clarify this in the revised caption.
>
> (b) Video Results: We plan to show the video results on the project pages; we have included some examples that can be found here:https://drive.google.com/drive/folders/1IUrSi-1VcBtr4FbIRrRSt9PuVrKYFE14?usp=sharing for immediate reference.
>
> (c) BANMo's Results: As discussed in our section A.5 (NeRF-based methods VS NMR-based methods), the NeRF-based methods such as ([1], [2], [3], [4], [6]) require many videos with from diverse viewpoints as input to obtain reasonable results. Given a single monocular video, they are not able to provide good results, which is also noted in CASA [11], Fig.6. And we will provide more results from NeRF-based methods in the paper.
>
> (d) We will provide the progression of the skeleton over iterations in the paper.
>
> We thank 2K9U for these invaluable suggestions and contributions to enhancing the quality of our work. We welcome any further feedback. We will incorporate all of the above changes and any further suggestions made during our discussion to revise our paper.
>
> [1] Noguchi, Atsuhiro, et al. "Watch it move: Unsupervised discovery of 3D joints for re-posing of articulated objects." Proceedings of the IEEE/CVF Conference on Computer Vision and Pattern Recognition. 2022.
>
> [2] Kuai, Tianshu, et al. "CAMM: Building Category-Agnostic and Animatable 3D Models from Monocular Videos." Proceedings of the IEEE/CVF Conference on Computer Vision and Pattern Recognition. 2023.
>
> [3] Yang G, Vo M, Neverova N, et al. “Banmo: Building animatable 3d neural models from many casual videos” Proceedings of the IEEE/CVF Conference on Computer Vision and Pattern Recognition. 2022.
>
> [4] Wu, Shangzhe, et al. "Magicpony: Learning articulated 3d animals in the wild." Proceedings of the IEEE/CVF Conference on Computer Vision and Pattern Recognition. 2023.
>
> [5] Yang, Gengshan, et al. "Lasr: Learning articulated shape reconstruction from a monocular video." Proceedings of the IEEE/CVF Conference on Computer Vision and Pattern Recognition. 2021.
>
> [6] Yang, Gengshan, et al. "Reconstructing animatable categories from videos." Proceedings of the IEEE/CVF Conference on Computer Vision and Pattern Recognition. 2023.
>
> [7] Yao, Chun-Han, et al. "Hi-lassie: High-fidelity articulated shape and skeleton discovery from sparse image ensemble." Proceedings of the IEEE/CVF Conference on Computer Vision and Pattern Recognition. 2023.
>
> [8] Xu, Zhan, et al. "Rignet: Neural rigging for articulated characters." arXiv preprint arXiv:2005.00559 (2020).
>
> [9] Xu, Zhan, et al. "Morig: Motion-aware rigging of character meshes from point clouds." SIGGRAPH Asia 2022 Conference Papers. 2022.
>
> [10] Yao, Chun-Han, et al. "Lassie: Learning articulated shapes from sparse image ensemble via 3d part discovery." Advances in Neural Information Processing Systems 35 (2022): 15296-15308.
>
> [11] Wu, Yuefan, et al. "Casa: Category-agnostic skeletal animal reconstruction." Advances in Neural Information Processing Systems 35 (2022): 28559-28574.

---

> > ### Comment · Reviewer_2K9U · 2023-11-21
> > **Rebuttal feedback**
> >
> > Dear authors,
> >
> > I appreciate the rebuttal. My main concerns haven't been addressed, listed as follows
> > - In general, I would like to see the direct edits in the revision instead of being promised in the comments. This applies both to the writing and figures/experiments. In its current form, it's difficult to get a global view of whether the paper passes the bar of acceptance. I would appreciate any effort made by the authors in this direction.
> > - Evaluation/comparison on the bone-discovery task. The provided notebook is not executable and indeed it returns an error when I tried it. I would suggest showing results with a figure instead of in a notebook which makes it extra complicated.
> > - Ablation on bone merging. I'm specifically referring to Sec 3.5 of WIM, where bones are merged based on their similarity in SE(3), as a connection cost. The current explanation appears confusing, and I don't understand why skinning weights and vertices play a role in the explanation.

---

> > > ### Author Response · Authors · 2023-11-22
> > > **Respond to Reviewer 2K9U -- Part 2**
> > >
> > > Certainly, we've updated the revised PDF and made it available for your review. In response to your insightful feedback, we've made several revisions:
> > >
> > > 1. We've modified certain terminologies, such as replacing 'implicit' with 'latent', and 'part assignment/decomposition' with 'skinning weights'. Additionally, 'inverse blend skinning' has been changed to 'blend skinning'.
> > > 2. We point out that the bone motion derived from the optical flow warp operation is particularly relevant in Section 3.3.
> > > 3. We've clarified that the rigidity coefficients are calculated from the skinning weights.
> > > 4. A comparison of our skeleton with RigNet is now included in Figure 4, along with a related discussion in Section 4.3.
> > > 5. We specify the differences between our approach and multiple existing methods in Table 2 and Section A.7.
> > > 6. A new discussion comparing our bone motion estimation method with WIM is added in Appendix A.6.
> > > 7. The introduction of NeRF-based and NMR-based methods has been relocated to the section on implementation details for better clarity and coherence.
> > >
> > > If you have any further questions or uncertainties, we are more than willing to provide additional explanations and present more experimental results to address them.

---

> > > > ### Comment · Reviewer_2K9U · 2023-11-22
> > > >
> > > > Thanks for putting effort into revising the paper and adding additional comparisons. As such, I updated my score to 6.
> > > >
> > > > The argument for 2D flow-based bone merging (in A.6) is still unconvincing. I would think merging bones based on their motion in 3D is more effective than doing so in 2D, which suffers from ambiguity in the depth dimension. Note this is orthogonal to whether the parts are being modeled in a semi-rigid or rigid manner.

---

> > > > > ### Author Response · Authors · 2023-11-22
> > > > > **Respond to Reviewer 2K9U -- Part 3**
> > > > >
> > > > > First of all, thank you for carefully reading our revised article!
> > > > >
> > > > > Additionally, thank you for raising the question: "What is the difference between using SE(3) and optical flow as the basis for judging bone motion?" We find this question very interesting and would like to discuss it further with you. We agree with your idea that using the SE(3) of each bone, which contains 3D information, intuitively should be better than using 2D optical flow (which lacks depth information). However, when we attempted to replace optical flow with SE(3), the results were not as good, especially in the initial stages. We analyze the reasons as follows: the SE(3) of the bones are learnable parameters, so the motion calculated for each part is accurate when the SE(3) predictions of the bones are very stable and accurate. This is why the bone merge stage in WIM is more like a post-processing step, where they first ensure the accurate learning of the mesh and the pose of each part before proceeding with the merge. In our method, the updating of the skeleton and other learnable parameters is done in synchrony, and at certain epochs, bones are merged or generated. The model needs to relearn the SE(3) and other parameters for the new bones, so in this case, using unstable SE(3) might not be as reliable as using the dependable 2D optical flow, which can be obtained by using existing models.

---

> > > > > > ### Author Response · Authors · 2023-11-23
> > > > > > **Respond to Reviewer 2K9U -- Part 4**
> > > > > >
> > > > > > We wish to further clarify our answer above to your interesting question about the use of SE(3) vs optical flow. (1) Note that from an arbitrary/general viewpoint, the motions of two bones across a joint are distinguishable because the (tangential) direction of motion in each bone with respect to rotation center at the meeting point of the bones (potential joint) is different. In such a case, both SE(3) and optical flow direction can be used to discriminate between the motions, and therefore allow/disallow merger of the bones. (2) From a degenerate viewpoint, say where the rotation of the bones is in the plane that projects as a line in the image plane, the directions of motions in the two bones across their meeting point (potential joint) are aligned and not distinguishable. However, such a degenerate case also disallows estimation of SE(3) from images anyway, and therefore neither SE(3) nor optical flow work. Therefore, for arbitrary views optical flow direction suffices, and SE(3) is not needed. Because of (1) the fact that we are not given SE(3), must estimate it, in fact we learn it, and the learned estimate is highly unstable at least in the early stages, and (2) that optical flow direction is reliably computable, we choose to use it. (In contrast, WIM separately estimates SE(3) and therefore they can and do use SE(3) for bone merging.)

---

### Official Review · Reviewer_gF7P · 2023-10-31

**Soundness:** 3 good
**Presentation:** 3 good
**Contribution:** 3 good
**Rating:** 8
**Confidence:** 4

**Summary:**

This paper focuses on learning non-rigid articulated objects from monocular videos. The proposed method employs a template surface mesh (explicit representation) and a skeleton (implicit representation) simultaneously without category-specific pretraining, followed by the proposed alternating optimization approach of surface mesh and skeleton. Additionally, a refinement method for the skeleton during optimization is introduced, allowing for adjusting the number of joints during the optimization. The method also introduces a part refinement technique, enhancing limb reconstruction. The proposed method is primarily compared with BANMo, and LASR, and evaluated across multiple benchmark datasets, demonstrating qualitative/quantitative improvements.

**Strengths:**

- The proposed method does not require category-specific pre-training, unlike MagicPony and BANMo.
- The proposed method demonstrates significant qualitative improvements over prior works.
- The proposed method incorporates a novel mechanism for adaptively learning the optimal number of skeleton joints.
- Extensive ablation studies are conducted.

**Weaknesses:**

**Major**
- Missing ablation of considering optical flow visibility (Eq. 4) and Laplacian contraction.
- The effectiveness of the rigidity coefficient/dynamic rigid does not seem substantial from Fig. 9 and Table 7.
- The most recent method, MagicPony, also employs implicit and explicit representation and strongly relates to the proposed work, yet a direct comparison in the experiment is missing.

**Minor**
- Although as a post-processing step, WIM [1] also infers the skeleton with a variable number of joints for articulated targets and the relation could be discussed.

[1] Noguchi et al. Watch It Move: Unsupervised Discovery of 3D Joints for Re-Posing of Articulated Objects. CVPR 2022.

**Questions:**

- How would the result change without considering optical flow visibility and Laplacian contraction?
- How should we interpret the improvement by rigid dynamics from Fig.9?
- Is there any reason more recent methods like CASA/MagicPony are not compared with the proposed work?

---

> ### Author Response · Authors · 2023-11-17
> **Respond to Reviewer gF7P -- Part 1(a)**
>
> We are thankful to gF7P for noting (1) the novelty of our adaptive learning mechanism; (2) the advantage of not requiring category-specific training that our method offers, (3) the improvements it makes over prior works; and (4) the extensive ablation studies we present.
>
> We greatly appreciate gF7P for the weaknesses noted and the questions they have raised. We outline below our response and how we plan to use these comments to improve the quality of our paper.
>
> (1) Ablation of considering optical flow visibility (Eq. 4) and Laplacian contraction: Thanks for pointing these out. We agree that this ablation will be useful. Our experiments reveal that removing visibility leads to instability and inaccuracy in calculating 2D bone motion when mapping 2D flow to 3D space. This is because typically each ray intersects the mesh at two points, with the 2D flow corresponding to the visible point. The non-visible point, often from a different part with an entirely different motion, causes instability when visibility is disregarded, resulting in an unreasonable skeleton. We will add this ablation to the main paper. We will also supplement it with ablation studies of the hyper-parameters of the mesh contraction module, including visual comparisons.
>
> (2) Effectiveness of Dynamic Rigidity (DR): The 2D key point transfer accuracy we measure is primarily from visible views, and this does not effectively assess the physical plausibility of the entire mesh. We agree that Figure 9 could be enhanced by including more views to show the mesh differences (e.g., the unreasonable depressions in the body in the 'without DR' experiment); accordingly, we will add more views and animal comparisons to better demonstrate the improvements brought by DR.
>
> (3) Comparison with MagicPony [4], WIM [1], CASA [8], and other Related Works: CAMM [2], and RAC [6] are similar as they all require ground truth skeletons (or pre-trained RigNet [9] to obtain skeleton) as input and leverage implicit fields (NeRF-based). As discussed in our Section A.5 (NeRF-based methods vs. NMR-based methods), such NeRF-based methods ([1],[2],[3],[4],[6]) place significant requirements on input videos in terms of the numbers of viewing angles and frame counts (over 1000 frames), as well as on ground truth camera poses (or use of pre-trained PoseNet [3] to obtain camera pose). When the input is a short monocular video, these methods fail to achieve satisfactory results (as seen from our Fig. 3 (b)). We will show more results from NeRF-based methods in the Appendix. In addition, we have found another recent work, Hi-lassie [7], which is similar to Magic Pony [4] but is based on NMR and yields acceptable results on monocular videos; our method, however, outperforms it as shown in https://drive.google.com/file/d/1VRdZeV0GK3pELwL8Y7u8OYHuXW9xWAoI/view. Further, we attempted to run CASA [8] but did not succeed in doing so since the authors did not release the ground truth skeleton for each animal, which is required to run the code as they use both predefined templates and skeletons. This makes a direct comparison difficult. We reached out to the authors multiple times via email but received no response.
>
> We will include a comparative discussion of all these methods in the paper.
>
> (4) How should we interpret the improvement by Dynamic Rigidity (DR): As shown in Figure 9, using ARAP terms leads to unnatural and uneven depressions in the reconstruction. We believe this is because the ARAP term uniformly encourages distance preservation between all connected vertices in the mesh, which is unreasonable because certain areas (junctions of different parts) should have the freedom to deform, and only distances within semi-rigid parts should be restricted. Changing ARAP to DR allows deformations to occur more at the junctions of parts rather than within semi-rigid parts, thus maintaining the shape integrity of each part. In contrast, using ARAP, deformations might occur within a part, resulting in unreasonable depressions or uneven shapes. We will add this explanation to the paper.

---

> ### Author Response · Authors · 2023-11-17
> **Respond to Reviewer gF7P -- Part 1(b)**
>
> [1] Noguchi, Atsuhiro, et al. "Watch it move: Unsupervised discovery of 3D joints for re-posing of articulated objects." Proceedings of the IEEE/CVF Conference on Computer Vision and Pattern Recognition. 2022.
>
> [2] Kuai, Tianshu, et al. "CAMM: Building Category-Agnostic and Animatable 3D Models from Monocular Videos." Proceedings of the IEEE/CVF Conference on Computer Vision and Pattern Recognition. 2023.
>
> [3] Yang G, Vo M, Neverova N, et al. “Banmo: Building animatable 3d neural models from many casual videos” Proceedings of the IEEE/CVF Conference on Computer Vision and Pattern Recognition. 2022.
>
> [4] Wu, Shangzhe, et al. "Magicpony: Learning articulated 3d animals in the wild." Proceedings of the IEEE/CVF Conference on Computer Vision and Pattern Recognition. 2023.
>
> [5] Yang, Gengshan, et al. "Lasr: Learning articulated shape reconstruction from a monocular video." Proceedings of the IEEE/CVF Conference on Computer Vision and Pattern Recognition. 2021.
>
> [6] Yang, Gengshan, et al. "Reconstructing animatable categories from videos." Proceedings of the IEEE/CVF Conference on Computer Vision and Pattern Recognition. 2023.
>
> [7] Yao, Chun-Han, et al. "Hi-lassie: High-fidelity articulated shape and skeleton discovery from sparse image ensemble." Proceedings of the IEEE/CVF Conference on Computer Vision and Pattern Recognition. 2023.
>
> [8] Wu, Yuefan, et al. "Casa: Category-agnostic skeletal animal reconstruction." Advances in Neural Information Processing Systems 35 (2022): 28559-28574.
>
> [9] Xu, Zhan, et al. "Rignet: Neural rigging for articulated characters." arXiv preprint arXiv:2005.00559 (2020).

---

> ### Comment · Reviewer_gF7P · 2023-11-22
> **Response to the rebuttal**
>
> I appreciate the authors for the updated manuscript with the additional visualizations and descriptions. The questions/weaknesses have been largely addressed except for the two items: ablation on flow visibility, and Laplacian contraction. I appreciate the authors’ effort in providing an additional explanation for the above items, however, it would be nicer to include the actual qualitative/quantitative results in the updated manuscript. As reviewer 2K9U also points out, it will make the paper stronger if a direct comparison of WIM on bone discovery is presented as well.

---

> > ### Author Response · Authors · 2023-11-22
> > **Respond to Reviewer gF7P -- Part 2**
> >
> > Thank you for taking the time to read our rebuttal!
> >
> > As you mentioned, two ablation experiments (Visibility and Laplacian contraction) are currently in progress (partly done).
> >
> > (1) We have already obtained results for key point transfer accuracy without visibility on five animal videos in the DAVIS dataset. The average is 77.3, which, although better than LASR (71.9) and ViSER (74.1), is lower than the results using visibility: 80.2. The PlanetZoo experiment is ongoing. The skeleton of DAVIS-camel without visibility is shown in the link: [https://drive.google.com/file/d/1a4HuUg-tJ8UF1_8c563tMQGZt9Fk1Trz/view?usp=sharing], which seems decent but not as good as using visibility.
> >
> > (2) By modifying parameters in Laplacian contraction, we found that the initial skeleton results are as shown in the link [https://drive.google.com/file/d/1UjaS_vkGyUx8QJj01OBluXdl-B6bo7QO/view?usp=sharing]. Although the number of bone points included is different, the overall structure is similar. We found out that all three types of initial skeletons can ultimately achieve results similar to those in the paper.
> >
> > We agree with your point that providing a comparison with WIM would make the paper stronger. We attempted to run WIM and compare it. However, the dataset we used does not have the ground truth camera pose, which is required in WIM. Additionally, our experiments focus on testing single monocular videos, while WIM is a method based on NeRF. We anticipate that without accurate camera poses and with a limited number of videos, it is highly likely that WIM may not be able to yield satisfactory results. In the alternative, we decided to compare with RigNet, which is trained with ground truth 3D supervision (mesh, skinning, and skeleton) to show the effectiveness of our method.
> >
> > We will update the manuscript as soon as possible after completing the additional ongoing experiments.

---

### Official Review · Reviewer_dR3E · 2023-10-31

**Soundness:** 2 fair
**Presentation:** 2 fair
**Contribution:** 2 fair
**Rating:** 6
**Confidence:** 3

**Summary:**

This paper proposes an algorithm to reconstruct a semi-nonrigid articulated object from RGB video. The geometry is modeled by an explicit mesh and the motion is modeled by a learnable skeleton and skinning. The geometry and motion models are jointly optimized on the video by flow supervision and several prior regularization losses. Experiments show that the proposed method can reconstruct animals and humans from the DAVIS dataset.

**Strengths:**

- The proposed method somehow worked on DAVIS to reconstruct Quadrupeds and the Human body.
- The idea of learning skinning and differentiable skeletons is interesting, this may inspire other deformation representations and general dynamic scene modeling. But not in this task (see weakness)

**Weaknesses:**

- The main concern lies in the necessity and value of the task in the current literature. There are two aspects to argue this:  1.) The reviewer guesses that given the current SoTA and technology in the community, the best way to model semi-nonrigid objects presented in this paper is to use template-based models. This paper presents animals and humans, which already have good template models. Only when the object motion structure differs a lot, and lacks a good template model, do we need some “unsupervised” algorithm to find the structure from videos. However, the paper never presents an example of such a case. 2.) The other way to recover the articulated object from the RGB (no depth) video is to first totally forget articulation and treat the scene as general dynamic 4D functions. In this way, given the current advanced dynamic rendering and reconstruction from monocular video, one can easily get the geometry as well as long-term correspondence first and then segment and easily extract the articulated object. Given these facts, I currently don’t believe this paper’s direction makes a real contribution. But as I write in the strengths, the idea of differentially learning the motion structure is interesting but needs to be carefully put into the right context and motivated nicely.
- The method is somehow complicated and lacks principles when presenting, making it not very easy to follow. Given a complex algorithm like this, I can hardly tell which component is really important, although some ablations are provided, given the complexity of the model, these may not be enough.
- The quality of the reconstruction shown in the figure is not appealing.

**Questions:**

See the weakness. The main question is the motivation for studying such a task in the way this paper presented.

---

> ### Author Response · Authors · 2023-11-17
> **Respond to Reviewer dR3E -- Part 1(a)**
>
> We are thankful to dR3E for noting the interestingness of the ideas of learning skinning and differentiable skeleton our method uses, and that these may inspire other representations and general dynamic scene modeling and greatly appreciate dR3E for the weaknesses noted and the questions they have raised. We outline below our responses and how we plan to use these comments to improve the quality of our paper.
>
> We disagree with the statement made by the reviewer that the model-based methods are the SOTA.
> Despite the use of predefined models in model-based methods that endow them with a wealth of prior knowledge, intuitively, one would expect their performance to surpass that of template-free methods. However, experimental results often contradict this assumption. For instance, references [2] (Fig 1) and [1] (Fig 6) illustrate that the outcomes of model-based approaches ([3],[4],[5],[6]) are suboptimal despite employing predefined templates. Several reasons account for this:
>
> 1. Model-based methods assume that the shape template is sufficiently accurate, significantly restrict the shape's deformability, and permit only simple transformations such as stretching the length of each bone. They do so because (1) when mesh deformation is unconstrained, the mesh often loses detail and transitions into a crude shape in the first several stages, resulting in the complete loss of the original prior information, and (2) even if the shape were allowed to undergo substantial changes, the overall scale or scale of each part might vary, potentially creating new parts, which prevent the mesh from aligning with the skeleton template. The adverse impact of this restriction is seen in recent works such as [12] and [11], where the results closely resemble the initial shape templates, focusing primarily on learning pose without reconstructing the unique detailed shapes of each instance. For example, in [12], since the video of a cow used a buffalo template, the resulting model still retained horns, and the A-CSM [5] reconstructed a camel with only one hump, despite the video featuring a two-humped camel (shown in [2] Fig 1)
>
> 2. Although there are templates for many animal species, individual variances, even within the same species, are vast. For example, the camel and horse templates used in A-CSM [5] and SMALify [13] differ greatly from actual camel instances, and often, these instance shapes do not exist within the deformation space allowed by these template-based methods. Hence, even with the correct pose, the rendered silhouette may significantly differ from the ground truth silhouette.
>
> 3. In real-world applications, out-of-distribution (OOD) objects are inevitable, and articulated objects are not limited to quadrupeds and humans. For instance, our tests (Table 1) included dinosaurs (Tyrannosaurus Rex) that cannot be accommodated by any existing quadruped/human templates. Such cases highlight the need for more flexible and adaptable reconstruction techniques.
>
> 2) Ignoring the articulation and directly learning general dynamic 4D functions often yields unsatisfactory results because such methods require a large number of videos, captured from multiple viewing angles, and ground truth camera poses. Also, it leads to a large number of parameters, entailing lengthy training periods (e.g., D-NeRF [7] needs over two days as it learns per-point motion without blend skinning), and making it difficult to converge (mentioned in [2]). Also, they rely on accurate camera pose (mentioned in [8], [9]) and have difficulties handling large motions (mentioned in [9]). They always leverage COLMAP to obtain the accurate camera pose, whose performance is known to be bad when the input video is short and the object has a large motion. These requirements are not met by our, more challenging scenario of a single monocular video without camera poses. Hence, directly getting the geometry and then long-term correspondence is not an option in our case.

---

> ### Author Response · Authors · 2023-11-17
> **Respond to Reviewer dR3E -- Part 1(b)**
>
> 3) Importances of Components: To the best of our understanding and experience, we feel that in a challenging task like the one we have attempted in this paper (single, monocular, short video without prior template), it is not easy to depend on one or a few sources of information to estimate the shape, joint structure, and articulated motion. Because of the highly compressed, convoluted, and time-varying interdependence of the sources, we think any estimation algorithm will have to make joint use of one or more of the information sources captured in our formulation and loss function, their relative contributions possibly varying from time to time. We therefore think that there is no single component in the framework that is consistently important to reach the end goal. This distributed nature of information is confirmed by the related previous papers [1-1], which also do not quantify the relative contributions of the different sources beyond presenting the ablations.  In our paper Sec. 4.3, we have presented the following ablation studies to demonstrate the effectiveness and analyze the reasons for the effectiveness of the various modules:
> (a) Physical Skeleton (ours) vs Virtual Bones
> (b) Refined Skeleton (ours) vs Initial Skeleton (Mesh Contraction)
> (c) Different Thresholds for Skeleton Refinement
> (d) Impact of Video Content on Skeleton
> (e) Efficacy of Dynamic Rigidity and Part Refinement
>
> Although Reviewers Z1Zw and gF7P have expressed satisfaction with the range of these ablations, we would be grateful if you (dR3E) could help us with which other ablations we could add; or how else could we specify module importances (given that there is no single consistently dominant module). We look forward to your suggestions about any additional experiments, analyses, etc. that we could include to improve our paper.
> In addition to a-e above, we agree with your suggestion and plan to provide ablation studies on different parameters of the mesh contraction module.
>
>
> 4) We would be interested to know in what ways we could improve the quality of reconstruction to make it more appealing. The results we have shown are analogous to those presented in SOTA papers; we have shown that our results are superior to those of the SOTA methods [2], under the same conditions, and even better than those of the methods that use additional information (e.g., Banmo [8], that uses DensePose to predict camera pose). Despite our challenging experimental setup (single short monocular video without any shape/skeleton prior and without ground truth camera), LIMR has shown impressive results (mesh and skeleton). We will provide analogous comparisons of our mesh results with a more recent SOTA method [10] at   https://drive.google.com/file/d/1VRdZeV0GK3pELwL8Y7u8OYHuXW9xWAoI/view; we will add these to the paper.

---

> ### Author Response · Authors · 2023-11-17
> **Respond to Reviewer dR3E -- Part 1(c)**
>
> [1] Tan, Jeff, Gengshan Yang, and Deva Ramanan. "Distilling Neural Fields for Real-Time Articulated Shape Reconstruction." Proceedings of the IEEE/CVF Conference on Computer Vision and Pattern Recognition. 2023.
>
> [2] Yang, Gengshan, et al. "Lasr: Learning articulated shape reconstruction from a monocular video." Proceedings of the IEEE/CVF Conference on Computer Vision and Pattern Recognition. 2021.
>
> [3] Rueegg, Nadine, et al. "Barc: Learning to regress 3d dog shape from images by exploiting breed information." Proceedings of the IEEE/CVF Conference on Computer Vision and Pattern Recognition. 2022.
>
> [4]Saito, Shunsuke, et al. "Pifuhd: Multi-level pixel-aligned implicit function for high-resolution 3d human digitization." Proceedings of the IEEE/CVF Conference on Computer Vision and Pattern Recognition. 2020.
>
> [5] Kulkarni, Nilesh, et al. "Articulation-aware canonical surface mapping." Proceedings of the IEEE/CVF Conference on Computer Vision and Pattern Recognition. 2020.
>
> [6] Kocabas, Muhammed, Nikos Athanasiou, and Michael J. Black. "Vibe: Video inference for human body pose and shape estimation." Proceedings of the IEEE/CVF conference on computer vision and pattern recognition. 2020.
>
> [7] Pumarola, Albert, et al. "D-nerf: Neural radiance fields for dynamic scenes." Proceedings of the IEEE/CVF Conference on Computer Vision and Pattern Recognition. 2021.
>
> [8] Weng, Chung-Yi, et al. "Humannerf: Free-viewpoint rendering of moving people from monocular video." Proceedings of the IEEE/CVF conference on computer vision and pattern Recognition. 2022.
>
> [9] Yang G, Vo M, Neverova N, et al. “Banmo: Building animatable 3d neural models from many casual videos” Proceedings of the IEEE/CVF Conference on Computer Vision and Pattern Recognition. 2022.
>
> [10] Yao, Chun-Han, et al. "Hi-lassie: High-fidelity articulated shape and skeleton discovery from sparse image ensemble." Proceedings of the IEEE/CVF Conference on Computer Vision and Pattern Recognition. 2023.
>
> [11] Goel, Shubham, et al. "Humans in 4D: Reconstructing and Tracking Humans with Transformers." arXiv preprint arXiv:2305.20091 (2023).
>
> [12] Wu, Yuefan, et al. "Casa: Category-agnostic skeletal animal reconstruction." Advances in Neural Information Processing Systems 35 (2022): 28559-28574.
>
> [13] Benjamin Biggs, Thomas Roddick, Andrew Fitzgibbon, and Roberto Cipolla. Creatures great and smal: Recovering the shape and motion of animals from video. In ACCV, pages 3–19. Springer, 2018.

---

> ### Comment · Reviewer_dR3E · 2023-11-21
> **Rebuttal comment**
>
> I appreciate the authors' effort in responding to each reviewer's comments with such detail. After reading both the comments and the authors' rebuttal, I am partially convinced that their approach makes sense under the context and setting of this paper. However, I suggest that the authors take the time to carefully clarify the novelty and motivation of their revision. Additionally, as other reviewers have pointed out, improving the presentation would be helpful in making the paper clearer. I have slightly raised my rating but have also lowered my confidence in the paper.

---

> > ### Author Response · Authors · 2023-11-22
> > **Respond to Reviewer dR3E -- Part 2**
> >
> > Thank you for your careful reading of our rebuttal. We will follow your and other reviewers' suggestions to improve the presentation of our paper, making it clearer.

---

### Official Review · Reviewer_Z1Zw · 2023-11-01

**Soundness:** 3 good
**Presentation:** 3 good
**Contribution:** 3 good
**Rating:** 8
**Confidence:** 4

**Summary:**

This paper aims to reconstruct the 3D shape of the moving articulated object from one or multiple monocular videos. The paper proposes LIMR (Learning Implicit Representation) that models both explicit information (3D shapes, colors, camera parameters) and implicit skeletal information. To iteratively estimate both implicit and explicit representations, the paper proposes Synergistic Iterative Optimization of Shape and Skeleton (SIOS2) algorithm that uses physical constraints as regularization terms. Experiments on standard datasets show that LIMR outperforms state-of-the-art category-agnostic methods.

**Strengths:**

1. This paper proposes a novel method for the important task of reconstructing moving articulated object from monocular videos. The proposed joint explicit and implicit representations seem effective in modeling both canonical structure and pose-dependent deformation.

2. Experiments have been conducted for a fair comparison with state-of-the-arts (i.e., LASR and BANMo) that do not take ground truth skeletons. The experiments include both qualitative (geometry and appearance) and quantitative (2D keypoint transfer and 3D shape reconstruction).

3. Extensive ablation studies have been conducted to show the importance of each component in Sec. 4.3.

4. Limitations and implementations have been discussed in detail. For example, one limitation is that the proposed method requires 10-20 hours to learn, which are comparable with baselines (LASR and BANMo on the order of a few hours).

**Weaknesses:**

1. More visualizations such as video comparisons like those shown in LASR and BANMo would be more intuitive and straightforward to show the object in the move/motion.

2. The proposed system includes multiple hyper-parameters and multiple separate but interdependent steps. Given the complications of the current system, it is unclear how robust this method is, eg, with regard to initialization, hyper-parameter settings, input video contents, etc.

**Questions:**

Some suggestions on the clarity of presentation:
1. Currently, the figures are not placed in an ascending order (Figure 7 before Figure 5) and the references to the figures in the text jump back and forth.

2. Some notifications are not well explained, eg, N_i in Eq. (3).

3. Typo: "(3)" in the first paragraph of appendix A: "quantitative"-->"qualitative".

---

> ### Author Response · Authors · 2023-11-17
> **Respond to Reviewer Z1Zw -- Part 1**
>
> We are thankful to Z1Zw for noting that: (1) our method effectively models both canonical structure and deformation. We conduct (2) a fair comparison with SOTA, (3) extensive ablation studies, and (4) discuss implementation and limitations in detail.
>
> We greatly appreciate Z1Zw for the weaknesses noted and the questions they have raised. We outline below our response and how we plan to use these comments to improve the quality of our paper.
>
> 1. Visual Comparisons: We will show videos of moving objects on the project page; we are including some examples here: https://drive.google.com/drive/folders/1IUrSi-1VcBtr4FbIRrRSt9PuVrKYFE14.
>
> 2. Hyperparameters and Robustness: The impact of video content and threshold hyperparameters related to skeleton updates on the reconstruction process is discussed in the experiments section. Further, we will: (a) elucidate the effects of skeleton initialization using different mesh construction parameters; (b) stability of results by comparing the results from multiple runs and the standard deviations; and (c) demonstrate the impact of input video contents by showing results for varying viewing directions.
>
> 3. Ordering of Figures and References: We acknowledge this weakness and will rectify it by renaming the figures.
>
> 4. Explaining Notation: We will add the clarification that \( N_i \) denotes the set of neighbor vertices of vertex \( i \), as well as look for any other such places for improvement.
>
> 5. Typographical Errors: Thank you for noting this typographical error. We will replace the incorrect reference to "quantitative" in the Appendix with "qualitative".

---

> > ### Comment · Reviewer_Z1Zw · 2023-11-22
> >
> > I appreciate the authors efforts in trying to address the concerns from all reviewers. I have read through all communications between the authors and the reviewers. My question about more video comparisons has been addressed.

---

> > > ### Author Response · Authors · 2023-11-22
> > >
> > > Thank you for taking the time to read through all the comments from reviewers and authors! We are very pleased to have addressed your questions.

---

### Author Response · Authors · 2023-11-22
**General Respond**

Thank you for every valuable suggestion from all the reviewers. Due to time constraints, some of our ablation studies are still ongoing. We have included the new experimental results and analyses that have been completed so far, and have revised the paper according to the reviewers' suggestions on presentation. The revised parts of the document have been highlighted in cyan color. We will update all new experimental results once the ablation studies are completed. Additionally, all video results and code will be released on our homepage and GitHub.

---

### Meta-Review · Area_Chair_6L5y · 2023-12-04

**Metareview:**

The submission proposes a new method for skeleton discovery from monocular videos with acceptable quality. The proposed method employs a template surface mesh (explicit representation) and a skeleton (implicit representation) simultaneously without category-specific pretraining, followed by the proposed alternating optimization approach of surface mesh and skeleton. Additionally, a refinement method for the skeleton during optimization is introduced, allowing for adjusting the number of joints during the optimization. The method also introduces a part refinement technique, enhancing limb reconstruction. The proposed method is primarily compared with BANMo, and LASR, and evaluated across multiple benchmark datasets, demonstrating qualitative/quantitative improvements.

Pros
*The proposed method does not require category-specific pre-training, unlike MagicPony and BANMo.
*The proposed method demonstrates significant qualitative improvements over prior works.
*The proposed method incorporates a novel mechanism for adaptively learning the optimal number of skeleton joints.
*Extensive ablation studies are conducted.

Cons
*Lack of comparison to prior work on motion-based skeleton discovery, especially WIM
*The proposed approach is hard to reproduced and follow up

**Justification For Why Not Higher Score:**

Two reviewers have concerns that the approach is hard to reproduce and follow up. Even reviewers who gave 8 don't think that the submission is wroth a spotlight.

**Justification For Why Not Lower Score:**

The submission proposed a new method for a challenging problems, with four out of five reviewers giving positive reviews.

---

### Decision · Program_Chairs · 2024-01-16

Accept (poster)